# CONGA: Confidence-and-Gradient-Aware Learning Rate Schedule for Test-Time Adaptation

**Shaoran Lv** [* 1]  **Xinyao Li** [* 1]  **Jingjing Li** [1 2]

## Abstract

Test-time adaptation (TTA) adapts pretrained models to test data on-the-fly. Current TTA methods have focused on *what* to adapt: lightweight domain-aware components (prompts, normalization statistics) updated with consistency-aware self-supervised losses. This work investigates the more fundamental yet underexplored optimization process, providing insights and guidelines on *how* to appropriately update models for TTA. By analyzing the optimization error during TTA, we identify a pivotal stability-plasticity trade-off: the model should adapt to novel distributions while retaining learned knowledge, which motivates our design of a CONfidence-and-Gradient-Aware scheduler (CONGA) to constrain model learning rate (LR) within an adaptive exploration interval. For each iteration, the lower bound encourages model exploration on informative confident samples, while the upper bound prevents aggressive overfitting to noisy optimization gradients. Based on our theoretical findings, an adaptation-progress-conditioned cosine decay function decides the specific LR within the interval. As an LR scheduler, CONGA is naturally applicable on existing TTA methods as a plug-in module, introducing little computation overheads. Extensive experiments and analysis demonstrate the superiority and validness of CONGA.

## 1. Introduction

Current deep models achieve superior performance under the assumption of being independent and identically distributed (i.i.d.), which is easily violated in real-world applications (Pan & Yang, 2009; Ben-David et al., 2010). To address this, transfer learning (TL) methods (Pan & Yang, 2009; Yosinski et al., 2014; Donahue et al., 2014; Li et al., 2024; 2025a;b) reuse the pretrained knowledge in the source model to handle target data under distribution shift. However, typical TL frameworks like pretrain-finetune require target training data in advance of deployment, which fails to handle scenarios with dynamically evolving test data, e.g., autonomous driving (Gong et al., 2022). To mitigate this, Test-Time Adaptation (TTA) emerges as a vital technique for adapting pretrained models to unseen domains during inference (Li et al., 2025c; Wang et al., 2020; Liang et al., 2020; Zhang et al., 2022). The outstanding challenge in TTA is to adapt with unlabeled test data streams on-the-fly, without access to the original source dataset.

Current TTA methods have extensively investigated *what* most critical model parts and target knowledge are to transfer in TTA. To meet the high efficiency requirements, lightweight domain-aware modules like normalization layers (Lim et al., 2023), input contexts (prompts) (Shu et al., 2022; Gan et al., 2023) or additional parameters (Li et al., 2025c) are adapted instead of the whole model parameters. To identify and learn from transferable data knowledge, the learnable modules are then optimized with confidence-aware losses (Wang et al., 2020) or consistency-aware self-supervised learning frameworks (Shu et al., 2022; Feng et al., 2023). Despite their effectiveness, these designs are typically implemented with default optimization configurations as supervised learning. Specifically, the constant learning rate (LR) or off-the-shelf LR schedule strategies might be challenged by the drastically changing data distribution and potentially harmful samples due to lack of reliable supervision (Yang et al., 2022). One aggressive update on unreliable data point could lead to drastic performance drop, while under-exploited target knowledge limits model capabilities.

To address this, this work explores *how* to effectively and safely implement current TTA methods by constraining their learning rates for each optimization step. To this end, we propose CONfidence-and-Gradient-Aware LR scheduler (CONGA) that adjusts LR at test time based on model out-

---

[*]Equal contribution [1]School of Computer Science and Engineering, University of Electronic Science and Technology of China, China [2]Yangtze Delta Region Institute (Huzhou), University of Electronic Science and Technology of China, Xisaishan Road, Huzhou, 313000, Zhejiang, China. Correspondence to: Jingjing Li <jjl@uestc.edu.cn>.

*Proceedings of the 43$^{rd}$ International Conference on Machine Learning*, Seoul, South Korea. PMLR 306, 2026. Copyright 2026 by the author(s).

puts, gradients and adaptation progress. We first revisit TTA by analyzing its optimization error as the adaptation progresses. For each iteration, the error is mainly bounded by two competing terms representing the error in previous iteration and variance of current gradient noise. The two terms are traded-off by LR, necessitating a time-varying learning rate strategy to achieve optimal optimization effects in different adaptation stages. Motivated by this, CONGA decides the LR in each iteration with three key designs. (1) To balance the optimization error and gradient noises, we apply a monotonically decaying strategy that enables smooth transition from early adaptation stages to late ones. Our analysis also motivates periodic restarts to maintain plasticity for non-stationary data streams. These strategies are adopted to select the optimal LR within an adaptive interval decided by a lower and upper bound. (2) The LR upper bound, termed Safety Gate, prevents large LR from unreliable updates. We improve LARS optimizer (You et al., 2017) by applying Logarithmic Damping to handle sudden gradient changes during the unstable and unpredictable TTA process. (3) The LR lower bound, termed the Learning Gate, encourages larger LR on samples without excessive noise. Our error reduction analysis reveals that the optimal update step is dominated by the gradient noise unreliability, motivating the Learning Gate to be inversely proportional to the noise for better model plasticity. Our main contributions are as follows:

- We revisit TTA from a more fundamental perspective of optimization process, revealing that the LR is not merely a hyperparameter but a critical factor for stable and reliable TTA. Our theoretical analysis of TTA optimization error demonstrates why standard LR schedulers fail and justifies the necessity of a confidence-and-gradient-aware design.

- We propose CONGA, an adaptive LR scheduler tailored for TTA. By dynamically stabilizing the optimization trajectory, it effectively mitigates catastrophic forgetting and gradient noise while introducing little computation overhead.

- CONGA is a plug-in module applicable to existing TTA baselines. Extensive experiments demonstrate that CONGA consistently enhances baseline performance across diverse benchmarks.

## 2. Related Work

**Test-time adaptation via adaptive objectives.** The paradigm of adapting models during inference has evolved significantly. Early foundational works like TTT (Sun et al., 2020) pioneered this by updating models via self-supervised tasks on test data, which necessitates modifying the source training process. Subsequently, Fully TTA (or source-free

TTA) emerged to adapt pre-trained models to arbitrary test domains using only unlabeled data streams without altering the source training. Existing TTA research predominantly revolves around the design of objective functions to guide domain alignment. Seminal works like Tent (Wang et al., 2020) utilize entropy minimization to update model parameters during inference. To mitigate the impact of noisy pseudo-labels, subsequent methods such as EATA (Niu et al., 2022) and DeYO (Lee et al., 2024) introduce sample filtering mechanisms based on entropy thresholds or temporal consistency. Focusing on representation structure, FOA (Niu et al., 2024) constrains the first-and second-order statistics of features before and after adaptation. In Continual Test-Time Adaptation (CTTA), methods like CoTTA (Wang et al., 2022) and Reservoir (Vray et al., 2025) emphasize long-term stability through Mean-Teacher architectures or memory buffers. Crucially, RoTTA (Yuan et al., 2023) addresses the stability-plasticity trade-off in dynamic scenarios using robust batch normalization and category-balanced memory banks. Despite their effectiveness, these objective-driven approaches largely treat the adaptation process as a black box, typically employing a constant learning rate. This overlooks the critical influence of optimization dynamics on TTA performance and stability.

**Test-time adaptation via optimization and gradient refinement.** Beyond designing loss functions, another line of research investigates the optimization trajectory and gradient quality to enhance stability. SAR (Niu et al., 2023) filters out noisy samples and utilizes sharpness-aware minimization to identify flat minima, which are more robust to pseudo-label noise. MGG (Deng et al., 2025) leverages meta-learning to generate adaptive gradients that are less sensitive to batch-specific variances. For learning rate control, PALM (Maharana et al., 2025) proposes a parameter-free adaptive learning rate method based on gradient and parameter norms, while SSA (Lee, 2025) achieves steady-state adaptation in dynamic environments by leveraging gradient-derived covariance for Bayesian weight enhancement. As a foundational context for adaptation, BN Adapt (Schneider et al., 2020) pioneered the online adaptation of Batch Normalization (BN) statistics to improve robustness against covariate shifts. Building on this concept of normalization and structural tuning, methods like NOTE (Gong et al., 2022) and EcoTTA (Song et al., 2023) focus on optimizing normalization statistics or lightweight visual prompts to stabilize the adaptation process in non-stationary environments. However, these methods often introduce additional meta-parameters or complex optimization processes. In contrast, our CONGA scheduler provides a lightweight solution to recalibrate optimization dynamics.

**Optimization dynamics and scheduling.** The design of learning rate schedule is fundamental to the convergence trajectory and generalization of deep neural networks. Clas-

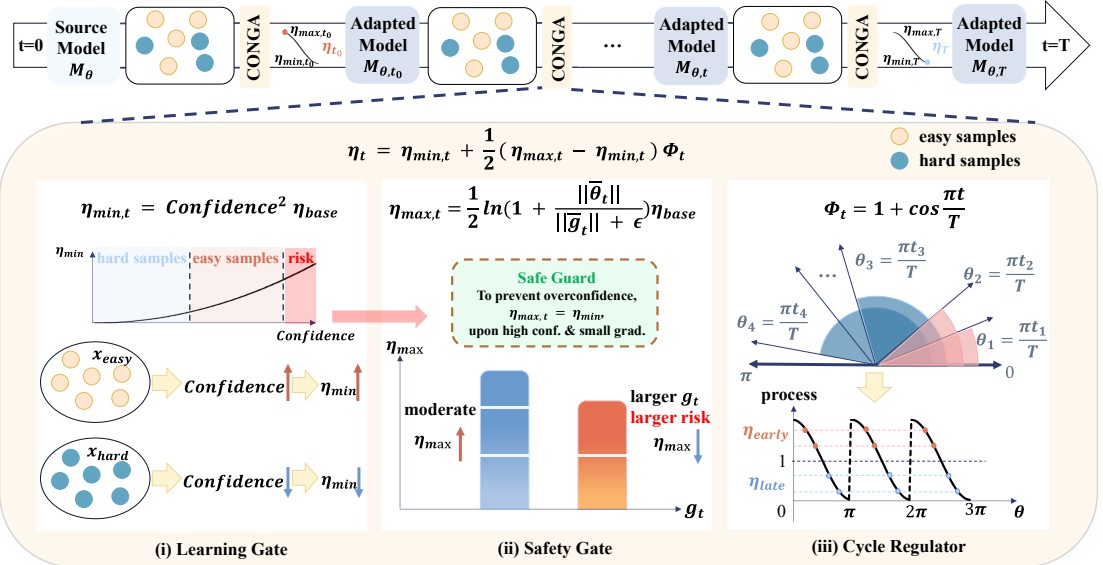

*Figure 1.* Overview of the proposed CONGA framework. The top arrow depicts the online adaptation process where the source model is refined on-the-fly during inference. Our framework modulates learning rate $\eta_t$ via three synergistic modules: (i) Learning Gate $\eta_{min}$ as confidence-aware LR lower bound to encourage updates based on prediction reliability; (ii) Safety Gate $\eta_{max}$ as gradient-aware LR upper bound to constrain updates on unreliable samples; (iii) Cycle Regulator for periodic cosine decay with restarts to regulate LR as the adaptation progresses.

sical strategies, such as Cosine Decay (Loshchilov & Hutter, 2016; Smith, 2017) and Linear Decay, have been widely adopted in supervised learning to escape sharp minima and reach stable optima. Beyond global scheduling, layer-wise adaptive rate methods such as LARS (You et al., 2017) adaptively calibrate the learning rate for each layer according to its respective gradient magnitudes. However, these techniques are primarily designed for static datasets with clean, ground-truth labels. In TTA, the optimization process is frequently disturbed by non-stationary data streams and noisy pseudo-labels, making traditional trust-region or fixed-decay methods sub-optimal. Our work bridges this gap by re-interpreting optimization dynamics through the lens of prediction confidence and gradient stability. By incorporating these TTA-specific metrics into a unified scheduling framework, we provide a principled way to maintain the balance between plasticity and stability without per-dataset hyperparameter tuning.

## 3. Method

### 3.1. Overview

We introduce CONGA, a framework that dynamically constrains the learning rate $\eta_t$ within a time-varying interval $[\eta_{min,t}, \eta_{max,t}]$, where $\eta_{max}$ and $\eta_{min}$ are adaptively derived from the base learning rate $\eta_{base}$. Fig. 1 presents CONGA framework with three components: A Cycle Regulator $\Phi(t)$ that divides the entire TTA process into cycles for restarts and decides the specific LR for each update. A

Safety Gate $\eta_{max}$ dependent on the update gradient to prevent overfitting on low-quality samples, reflected by steep gradient. A Learning Gate $\eta_{min}$ for stable updates on high-quality prediction, reflected by high softmax confidence.

**Setup and notation.** Consider a pre-trained model $M_\theta$ deployed to an unseen target domain. During inference, the model receives a continuous stream of unlabeled target samples $\{x_t\}_{t=1}^T$. For each incoming batch $x_t$, the model parameters $\theta_t$ are updated to overcome potential distribution shifts. We formalize the adaptation process as updating model parameters to fit the dynamically changing target distribution: let $w_t$ denote the theoretically optimal parameters that best describe the target distribution at time $t$. The goal is to update $\theta_t$ such that it aligns with the target $w_t$ as closely as possible in a non-stationary environment.

### 3.2. Cycle Regulator: A Periodic Decay Scheduler

The adaptation process can be formalized as "tracking" the optimal parameters to fit the target distribution at each time step $t$. We first provide theoretical understanding of the *tracking error* during TTA.

**Theorem 3.1** (Dynamics of Tracking Error). *Let $e_t \triangleq \theta_t - w_t$ denote the tracking error at time $t$. Under the assumption of a stochastic update with learning rate $\eta_t$ and a target drift magnitude $\|\delta_t\| = \|w_{t+1} - w_t\| \leq \delta$, suppose the stochastic gradient noise $\kappa_t$ admits a decomposition $\kappa_t = b_t + \xi_t$, where stochastic noise $\xi_t$ satisfies $\mathbb{E}[\xi_t] = 0$, $\mathbb{E}[\|\xi_t\|^2] = \sigma_t^2$, and the bias term is bounded as $\|b_t\| \leq B$. Then, the expectation of the Mean Squared Error at time $t+1$ satisfies*

*Table 1.* Comparison of different LR decay schedules in TTA based on time derivatives ($\dot{\eta}$).

| Schedule | Functional Form $\eta_t$ | Derivative at $0$ ($\dot{\eta}_0$) | Derivative at $T$ ($\dot{\eta}_T$) |
|---|---|---|---|
| Step | $\eta_{max} \cdot \gamma^{\lfloor t/s \rfloor}$ | 0 (Locally) | Discontinuous |
| Linear | $\eta_{max} - k \cdot t$ | Constant $< 0$ | Constant $< 0$ |
| Exponential | $\eta_{max} \cdot \gamma^t$ | Large $< 0$ | Small $< 0$ |
| **Cosine (Ours)** | $\eta_{min} + \frac{1}{2}(\eta_\Delta)(1 + \cos(\frac{t\pi}{T}))$ | **Zero** | **Zero** |

*the following recurrence relation:*

$$\mathbb{E}[\|e_{t+1}\|^2] \leq \underbrace{(1 - \eta_t)\mathbb{E}[\|e_t\|^2]}_{\text{Error Contraction}} + \underbrace{\eta_t^2 \sigma_t^2}_{\text{Noise Variance}}$$

$$+ \underbrace{\mathcal{O}(\eta_t B^2 + \frac{\delta^2}{\eta_t})}_{\text{Structural Drift}} \qquad (1)$$

**Implications of error contraction and noise variance.**
Theorem 3.1 (The proof is provided in Sec. A.1.) elucidates a fundamental trade-off in TTA, characterized by the competition between the Error Contraction, which quantifies the reduction in the previous tracking error enabled by the current gradient-based adaptation, and the Stochastic Variance, which captures the cumulative uncertainty introduced by stochastic gradient noise during the adaptation process. Specifically, the adaptation process is governed by a transition from an error-dominant regime to a noise-dominant regime: in the early stages, the error is dominated by the initial alignment error $\mathbb{E}[\|e_t\|^2]$, which indicates a large domain shift and requires a larger LR for quick alignment; conversely, in the late stages, the domain shift error vanishes and the variance of the stochastic noise $\eta_t^2 \sigma_t^2$ dominates, necessitating $\eta_t \to 0$. This derivation theoretically confirms that the ideal schedule for TTA should begin with high LR to adapt to the domain shift and gradually decay to suppress noise in later stages.

**Implications of structural drift.** Although the decay schedule progressively diminishes the error contraction and noise terms, it fails to resolve the structural drift, which denotes the constant deviation from the optimal target manifold driven by the coupling of target drift $\delta$ and gradient bias $B$. As a result, this drift remains persistent and accumulates over time. This accumulation eventually causes the model to fall into suboptimal local traps. Therefore, a periodic restart is necessary to escape such traps.

**Cosine decay strategy.** While the necessity of decay and restart is established, the functional form of decay is critical. We argue that an ideal TTA schedule should satisfy three physical constraints: smoothness to avoid momentum instability, a sustained initial impulse to rapidly correct the large initial domain shift, and a "soft landing" during the late stages of adaptation to minimize parameter variance. Tab. 1 compares various candidate LR decay strategies. Step decay

suffers from discontinuities, while Linear and Exponential strategies decay too rapidly at the beginning and reach a non-zero terminal LR at the end, leading to insufficient adaptation and late-stage instability. In contrast, Cosine decay satisfies all established requirements. Guided by the theoretical requirements for rapid adaptation and noise suppression, we define the Cycle Regulator $\Phi(t)$ based on a cosine decay function:

$$\Phi(t) = 1 + \cos\left(\frac{\pi t}{T_i}\right) \qquad (2)$$

where $T_i$ is a hyperparameter representing the length of the $i$-th cycle and $t$ is the current step within the cycle. Eq. (2) defines the LR decay process and determines restarts of the adaptation process.

### 3.3. Safety Gate

To constrain the model optimization step within a safe radius, we define the Safety Gate $\eta_{max,t}$ as the base learning rate $\eta_{base}$ scaled by a time-varying scaling factor $\mathcal{S}_{max,t}$.

Our design draws inspiration from the LARS optimizer (You et al., 2017), which scales the learning rate based on the Trust Region ratio ($\|\theta\|/\|g\|$). Formally, this ratio defines a local trust radius that constrains the update magnitude relative to the current scale of the parameters. While effective for supervised training, this linear scaling does not suit the TTA setting in its current form. TTA relies on unreliable gradients derived from out-of-distribution unlabeled data, which potentially mislead the model updates and hinder effective adaptation. We investigate such unreliable adaptation empirically in Fig. 2. As illustrated by the gray trajectory, standard linear scaling factor exhibits severe high-frequency fluctuation due to the noisy TTA gradients. Such optimization shocks can destabilize the model, causing it to diverge from the pre-trained manifold.

To address this, we propose a logarithmic mechanism that preserves the original information while suppressing high-variance noise. As shown by the red trajectory in Fig. 2, this logarithmic mapping significantly smooths the scaling factor, ensuring a robust adaptation process. Formally, we define the Safety Gate $\eta_{max,t}$ as:

$$\eta_{max,t} = \eta_{base} \cdot \underbrace{\frac{1}{2} \ln\left(1 + \frac{\|\bar{\theta}_t\|}{\|\bar{g}_t\| + \epsilon}\right)}_{\mathcal{S}_{max,t}}, \qquad (3)$$

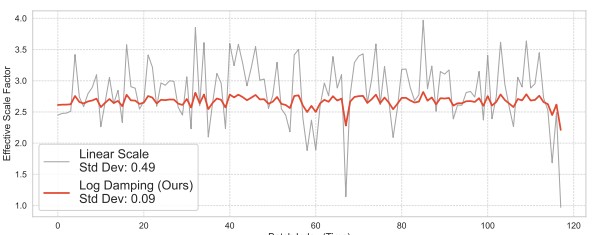

*Figure 2.* Impact of Logarithmic Damping on the linear scaling factor. We visualize the evolution of the dynamic scaling multiplier $\mathcal{S}_{max,t}$ on ImageNet-A. The plot compares our Logarithmic factor (Red) against a standard Linear factor derived from LARS (Gray). The linear scaling suffers from severe high-frequency fluctuation due to noisy gradients, whereas our logarithmic design produces a stable, dampened modulation trajectory, preventing catastrophic forgetting induced by unreliable gradient.

where $\|\bar{\theta}_t\|$ and $\|\bar{g}_t\|$ are the exponential moving average (EMA) of parameter and gradient norms, respectively. $\eta_{max,t}$ acts as a soft constraint, capping the LR step based on the logarithmic scale of the parameter-to-gradient ratio.

To prevent overconfidence, we implement a **Safe Guard** mechanism that triggers when $\eta_{max,t} > 3 \cdot \eta_{base}$ and $\rho_t > 0.75$, where $\rho_t$ is defined as the mean of the maximum softmax probabilities across all samples in the batch. This state indicates that the error contraction term has been sufficiently minimized, and aggressive updates may induce catastrophic forgetting. When triggered, the Safe Guard resets $\eta_t$ to $\eta_{min,t}$, shifting the focus from rapid adaptation to late-stage stabilization.

### 3.4. Learning Gate

**Theorem 3.2** (Optimal Learning Rate for Error Reduction). *Let $\Delta\mathcal{E}_t \triangleq \mathbb{E}[\|e_t\|^2] - \mathbb{E}[\|e_{t+1}\|^2]$ be the expected error reduction. Under the recurrence relation in Theorem 3.1, the optimal learning rate $\eta_t^*$ that maximizes the lower bound of $\Delta\mathcal{E}_t$ satisfies:*

$$\eta_t^* \propto 1/\sigma_t^2, \tag{4}$$

Theorem 3.2 (The proof is provided in Sec. A.2.) reveals that the optimal step size is primarily governed by the inverse of the stochastic gradient noise variance $\sigma_t^2$. This suggests a fundamental trade-off in TTA: while larger updates accelerate error contraction, they concurrently amplify the risk of being misled by stochastic noise. Consequently, when the gradient signal is reliable (i.e., low $\sigma_t^2$), a larger learning rate is theoretically preferred to maximize plasticity.

To bridge this theoretical optimum with an actionable constraint, we argue that the *Learning Gate* $\eta_{min,t}$ should track the trend of $\eta_t^*$ to prevent the model from entering an under-exploited state during the learning rate decay process. Specifically, we enforce a plasticity floor that adheres to the derived relationship $\eta_{min,t} \propto 1/\sigma_t^2$, ensuring that the lower bound of the adaptation rate remains matching with

the potential optimization gain.

However, in practical TTA environments, the noise variance $\sigma_t^2$ is not directly observable. To address this, we leverage predictive confidence as an empirical proxy. Following the established principle in importance sampling (Katharopoulos & Fleuret, 2018) which asserts that gradient noise is inversely related to prediction confidence, we assume the relationship $1/\sigma_t^2 \propto \bar{C}_t^2$. Here, $\bar{C}_t$ represents the reliable sample density, quantified as:

$$\bar{C}_t = \frac{1}{B}\sum_{i=1}^{B} \mathbb{I}(\max p(y|x_i) > \tau_C), \tag{5}$$

where $B$ is the batch size and $\tau_C$ is set to 0.9.

By substituting this proxy into the optimization constraint, we define the *Learning Gate* $\eta_{min,t}$ as:

$$\eta_{min,t} = \eta_{base} \cdot \bar{C}_t^2. \tag{6}$$

This formulation effectively lifts the learning rate floor toward the optimal trajectory $\eta_t^*$ when high-quality samples are encountered, thereby maximizing exploitation while maintaining a safety margin against low-confidence and noisy updates.

### 3.5. Overall Formulation

Combining the Cycle Regulator in Eq. (2), the Safety Gate in Eq. (3) and the Learning Gate in Eq. (6), we can formalize the schedule rule of CONGA at step $t$ within a cycle $T_i$ as:

$$\eta_t = \eta_{min,t} + \frac{1}{2}(\eta_{max,t} - \eta_{min,t})\Phi(t). \tag{7}$$

An overall algorithm for CONGA is shown in Algorithm 1.

## 4. Experiments

### 4.1. Experimental Setup

**Datasets and models.** We utilize the standard **ImageNet-C** benchmark (Hendrycks & Dietterich, 2019) for corruption robustness with both **ResNet-50-BN** (He et al., 2016) and **ViT-Base** (Dosovitskiy, 2020) backbones. To further assess generalization on optimization-sensitive architectures, we extend the evaluation on ViT-Base to benchmarks exhibiting severe distribution shifts: **ImageNet-A** (Hendrycks et al., 2021b), **ImageNet-R** (Hendrycks et al., 2021a), and **ImageNet-Sketch** (Wang et al., 2019).

**Baselines.** We evaluate our proposed method by categorizing existing TTA baselines into two groups. The first group focuses on adaptation objectives and data-centric strategies that are inherently orthogonal to our approach. These include foundational entropy minimization **Tent** (Wang et al., 2020), sample-filtering-based objectives **DeYO** (Lee

*Table 2.* Average accuracy (%) under ImageNet-C (level 5) in the standard TTA setting. Best results are bolded. Results of optimization-based baselines are highlighted with a red background.

| Method | Noise | | | Blur | | | | Weather | | | | Digit | | | | Avg | |
|---|---|---|---|---|---|---|---|---|---|---|---|---|---|---|---|---|---|
| | Gau. | Shot | Imp. | Def. | Glass | Motion | Zoom | Snow | Frost | Fog | Bright | Contrast | Elastic | Pixel | JPEG | Avg | Δ |
| **ResNet-50-BN base** | | | | | | | | | | | | | | | | | |
| NoAdapt | 2.2 | 2.9 | 1.8 | 17.9 | 9.8 | 14.8 | 22.5 | 16.9 | 23.3 | 24.4 | 58.9 | 5.4 | 16.9 | 20.7 | 31.7 | 18.0 | |
| SAR | 30.9 | 33.3 | 31.5 | 27.5 | 27.3 | 42.2 | 49.5 | 47.8 | 42.5 | 57.3 | 67.4 | 38.1 | 54.6 | 58.7 | 52.3 | 44.1 | |
| Tent | 29.9 | 32.0 | 31.3 | 28.0 | 27.4 | 41.3 | 49.2 | 47.0 | 41.4 | 57.7 | 67.5 | 28.0 | 54.8 | 58.6 | 52.7 | 43.2 | |
| + Ours | 34.2 | 36.1 | 35.5 | 31.1 | 31.2 | 46.0 | 51.7 | 50.9 | 43.2 | 59.4 | 68.0 | 27.9 | 57.6 | 60.6 | 55.0 | 45.9 | +2.7$_{\pm0.07}$ |
| EATA | 35.6 | 37.4 | 36.6 | 33.8 | 33.1 | 47.0 | 52.7 | 51.4 | 45.5 | 59.9 | 68.2 | 44.3 | 58.0 | 60.5 | 55.2 | 47.9 | |
| + Ours | 35.6 | 38.2 | 36.9 | 33.5 | 33.7 | 47.7 | 52.8 | 52.2 | 45.7 | 60.3 | 67.8 | 44.5 | 58.3 | 60.7 | 55.2 | 48.2 | +0.3$_{\pm0.13}$ |
| Deyo | 35.5 | **38.6** | **38.1** | **33.3** | 32.8 | **48.5** | **52.9** | 52.4 | 46.2 | 60.5 | **68.1** | 41.5 | 58.4 | **61.4** | 55.7 | 48.3 | |
| + Ours | **36.1** | 38.5 | 37.5 | 33.2 | **33.0** | 48.5 | 52.9 | **52.8** | **46.3** | **60.6** | 67.5 | 45.8 | 58.6 | 61.3 | 55.9 | 48.6 | +0.3$_{\pm0.05}$ |
| **ViT base** | | | | | | | | | | | | | | | | | |
| NoAdapt | 56.8 | 56.8 | 57.5 | 46.9 | 35.6 | 53.1 | 44.8 | 62.2 | 62.5 | 65.7 | 77.7 | 32.6 | 46.0 | 67.0 | 67.6 | 55.5 | |
| LAME | 56.5 | 56.6 | 57.3 | 46.4 | 34.8 | 52.7 | 44.3 | 58.4 | 61.6 | 63.1 | 77.5 | 24.7 | 44.6 | 66.6 | 67.3 | 54.2 | |
| T3A | 56.4 | 56.9 | 57.3 | 47.8 | 37.7 | 54.2 | 46.9 | 63.6 | 60.8 | 68.5 | 78.1 | 38.4 | 50.0 | 67.6 | 69.0 | 56.9 | |
| SAR | 59.2 | 60.5 | 60.7 | 57.5 | 55.6 | 61.8 | 57.6 | 65.9 | 63.5 | 69.1 | 78.7 | 45.7 | 62.4 | 71.9 | 70.3 | 62.7 | |
| Tent | 60.3 | 61.6 | 61.8 | 59.2 | 56.5 | 63.5 | 59.2 | 61.5 | 64.5 | 16.6 | 79.3 | 67.5 | 61.3 | 72.6 | 70.7 | 61.1 | |
| + Ours | 61.1 | 62.7 | 62.8 | 60.3 | 59.0 | 65.0 | 61.5 | 62.9 | 64.1 | 44.6 | 79.8 | 68.7 | 65.2 | 74.2 | 72.1 | 64.3 | +3.2$_{\pm1.09}$ |
| FOA | 60.9 | 61.6 | 62.7 | 56.5 | 49.8 | 60.8 | 56.4 | 66.6 | 63.3 | 69.2 | 79.4 | 64.3 | 58.2 | 71.4 | 70.3 | 63.4 | |
| + Ours | 61.2 | 62.0 | 62.9 | 56.8 | 51.0 | 61.4 | 57.2 | 67.1 | 63.8 | 69.8 | 79.7 | 64.6 | 59.5 | 72.1 | 70.9 | 64.0 | +0.6$_{\pm0.09}$ |
| Deyo | 62.7 | 64.1 | 63.8 | 60.2 | 60.7 | 66.5 | 62.9 | 70.9 | 69.6 | 73.1 | **80.6** | 38.3 | 69.6 | 75.7 | 73.7 | 66.2 | |
| + Ours | **62.9** | **64.3** | **64.1** | **60.7** | **62.2** | **67.1** | **64.3** | **71.9** | **70.4** | **73.2** | 80.6 | **62.9** | **72.1** | **76.6** | **74.4** | **68.5** | +2.3$_{\pm1.03}$ |

et al., 2024), **EATA** (Niu et al., 2022), and regularization **FOA** (Niu et al., 2024). We also incorporate **Reservoir** (Vray et al., 2025), a data-stream management strategy that utilizes domain-aware memory buffering for Continual TTA (CTTA). The second group consists of adaptation mechanisms and optimizers that are non-orthogonal to our approach, serving as direct counterparts. These include prototype-based or optimization-free inference mechanisms **T3A** (Iwasawa & Matsuo, 2021) and **LAME** (Boudiaf et al., 2022), as well as optimization-based schemes including the sharpness-aware minimizer **SAR** (Niu et al., 2023) and the layer-adaptive learning rate strategy **PALM** (Maharana et al., 2025). Detailed descriptions of baselines can be found in Sec. A.3.

**Implementation details.** Our method is implemented in PyTorch, strictly following baseline protocols. We set the base learning rate $\eta_{base}$ to $1 \times 10^{-3}$ for ViT-TTA, $2 \times 10^{-3}$ for ResNet, and $2.5 \times 10^{-4}$ for ViT-CTTA. To compute $\eta_{max}$, we track norms via an EMA with fixed $\alpha = 0.99$. Regarding restarts, we adopt a fixed task-dependent policy: resetting every 100 batches. We employ the standard SGD optimizer (Bottou, 2010) to more rigorously isolate and demonstrate the intrinsic effectiveness of our proposed schedule. Unless otherwise specified, all standard deviations are calculated across 5 independent runs. More implementation details and hyperparameter choices are presented in Sec. B.

## 4.2. Main Results

**Results on ImageNet-C in the TTA setting.** As shown in Tab. 2, our proposed CONGA consistently outperforms all baseline methods across various corruption types. Notably, it achieves a significant gain of **+3.2%** on ViT-backbone and **+2.7%** on ResNet-backbone when integrated with Tent. These enhancements enable the augmented Tent to surpass SAR (44.1% and 62.7%, respectively), a strong optimization-based baseline. This consistent improvement across different architectures and loss functions validates that CONGA acts as an orthogonal plug-in that effectively revitalizes the optimization dynamics of TTA.

**Results on ImageNet-C in the CTTA setting.** Tab. 3 reports the results under the more challenging CTTA setting over 10 recurring cycles. Our method consistently yields improvements, notably achieving a **+0.6%** gain over the state-of-the-art Reservoir baseline on ViT-Base. Given that CTTA is prone to catastrophic collapse due to error accumulation, these results underscore the long-term stabilization provided by our mechanism. While PALM achieves competitive accuracy, it introduces $2\times$ BP computational overhead. In contrast, CONGA maintains a strictly $1\times$BP overhead with near-zero memory growth, achieving superior performance-efficiency trade-offs as detailed in Sec. D.4.

**Results on other ImageNet variants.** Tab. 4 extends our evaluation to domains exhibiting severe distribution shifts (ImageNet-A, -Sketch, and -R). Our method consistently enhances performance across these diverse domains, demon-

*Table 3.* Average accuracy (%) under ImageNet-C (level 5) in the CTTA setting. We report the performance at the final round of 10 recurring visits. Best results are bolded. Results of optimization-based baselines are highlighted with a red background.

| | Noise | | | Blur | | | | Weather | | | | Digit | | | | Avg | |
| Method | Gau. | Shot | Imp. | Def. | Glass | Motion | Zoom | Snow | Frost | Fog | Bright | Contrast | Elastic | Pixel | JPEG | Avg | $\Delta$ |
|---|---|---|---|---|---|---|---|---|---|---|---|---|---|---|---|---|---|
| **ResNet-50-BN base** | | | | | | | | | | | | | | | | | |
| NoAdapt | 15.0 | 15.8 | 14.8 | 14.9 | 15.4 | 26.2 | 38.5 | 34.4 | 31.8 | 48.0 | 65.0 | 16.7 | 43.8 | 48.7 | 40.1 | 31.3 | |
| PALM | 37.4 | 39.0 | 38.1 | 32.2 | 32.1 | 37.0 | 40.7 | 41.5 | 36.3 | 42.8 | 51.8 | 30.5 | 44.9 | 47.3 | 44.6 | 41.2 | |
| EATA | **28.1** | 31.7 | **31.5** | **26.6** | 28.1 | 37.9 | 47.7 | 43.5 | **40.7** | 54.9 | 66.5 | 32.2 | 52.1 | 55.5 | **50.3** | 41.8 | |
| **+ Ours** | **28.1** | **31.8** | 31.4 | 26.0 | **28.7** | **38.8** | **48.3** | **45.2** | 40.2 | **55.3** | **66.6** | **34.0** | **52.7** | **55.9** | 50.2 | **42.2** | +0.4$_{\pm 0.06}$ |
| **ViT base** | | | | | | | | | | | | | | | | | |
| NoAdapt | 57.3 | 58.9 | 58.6 | 41.7 | 31.4 | 49.2 | 40.5 | 62.5 | 61.2 | 42.4 | 77.4 | 13.0 | 43.9 | 65.5 | 66.6 | 51.3 | |
| PALM | 62.1 | 64.4 | 64.0 | 56.1 | 58.3 | 63.3 | 53.4 | 67.0 | 61.9 | 68.0 | 78.1 | 61.8 | 64.2 | 72.2 | 71.4 | 64.4 | |
| Tent | 61.3 | 62.7 | 62.2 | 55.1 | 50.6 | 61.3 | 54.6 | 64.1 | 61.5 | 68.1 | 79.1 | 62.0 | 53.0 | 70.6 | 70.5 | 62.4 | |
| **+ Ours** | 61.2 | 62.7 | 62.2 | 54.9 | 50.7 | 60.9 | 55.2 | 63.9 | 61.8 | 67.9 | 79.0 | 61.4 | 55.0 | 71.3 | 70.4 | 62.6 | +0.2$_{\pm 0.00}$ |
| Reservoir | 63.3 | 64.4 | 63.3 | 58.7 | 58.7 | 62.7 | 59.8 | 68.2 | 68.2 | 72.2 | **79.2** | **65.1** | 62.3 | 72.1 | 72.5 | 66.0 | |
| **+ Ours** | **63.8** | **65.2** | **63.9** | **59.5** | **59.2** | **63.6** | **60.9** | **68.5** | **68.8** | **72.6** | **79.2** | 64.6 | **63.7** | **72.8** | **72.8** | **66.6** | +0.6$_{\pm 0.03}$ |

*Figure 3.* Ablation study on decay schedule and adaptive LR bounds. (a) Performance comparison of CONGA with decaying LR (red line) and LR warmups (Goyal et al., 2017) (blue line) on ImageNet-Sketch, with baseline method FOA. (b) Sensitivity analysis of $\eta_{max}$ for Tent and DeYO on ImageNet-A. (c) Sensitivity analysis of $\eta_{min}$ for Tent on ImageNet-A (Low conf.) and ImageNet-R (High conf.). The base learning rate $\eta_{base}$ is set to $10^{-3}$. High conf. and Low conf. denotes higher confidence data and lower confidence data, respectively. Star markers indicate the adaptive $\eta_{max}$ and $\eta_{min}$ (averaged across batches) automatically determined by our CONGA, along with their corresponding accuracy gains.

*Table 4.* Average TTA classification accuracy (%) in different ImageNet domains.

| Method | A | $\Delta$ | Sketch | $\Delta$ | R | $\Delta$ |
|---|---|---|---|---|---|---|
| NoAdapt | 0.1 | | 44.9 | | 59.5 | |
| FOA | 50.9 | | 46.3 | | 60.3 | |
| **+ Ours** | 51.3 | +0.4$_{\pm 0.05}$ | 47.1 | +0.8$_{\pm 0.03}$ | 61.1 | +0.8$_{\pm 0.17}$ |
| TENT | 52.9 | | 49.1 | | 63.9 | |
| **+ Ours** | 53.6 | +0.7$_{\pm 0.04}$ | 50.7 | +1.6$_{\pm 0.11}$ | 65.2 | +1.3$_{\pm 0.05}$ |
| Deyo | 55.2 | | 52.2 | | 66.1 | |
| **+ Ours** | 55.5 | +0.3$_{\pm 0.03}$ | 53.0 | +0.8$_{\pm 0.10}$ | 68.3 | +2.2$_{\pm 0.19}$ |

*Table 5.* Comparison of different LR decay strategies based on Tent on ViT. "Baseline" adopts constant LR of 1e-3.

| Dataset | Baseline | Step | Linear | Exp | Cosine |
|---|---|---|---|---|---|
| ImageNet-Sketch | 49.1 | 48.8 | 49.5 | 48.4 | **50.1** |
| ImageNet-A | 52.9 | 52.8 | 53.2 | 52.3 | **53.6** |

### 4.3. Ablation Study

**Decay schedule ablation.** We first evaluate the impact of the learning rate decay strategy on the ImageNet-Sketch benchmark. As illustrated in Fig. 3(a), the decay schedule consistently outperforms the warmup schedule, exhibiting a faster convergence rate and achieving a higher accuracy ceiling. This suggests that for test-time adaptation, a higher initial learning rate is beneficial for rapidly capturing domain-specific features, while subsequent decay ensures stability. Furthermore, Fig. 4 demonstrates the critical role of periodic restarts that prevent the optimization from stalling in suboptimal local minima.

**Adaptive $\eta_{max}$ and $\eta_{min}$ ablation.** Fig. 3(b) investigates

strating robustness under more challenging adaptation scenarios. Our approach achieves substantial gains when integrated with stronger baselines like DeYO (Avg. **+1.1%**). This observation suggests that our method effectively leverages the optimization potential of capable baselines to tackle complex, real-world domain generalizations. We provide per-cycle results and standard deviations in Sec. D.1.

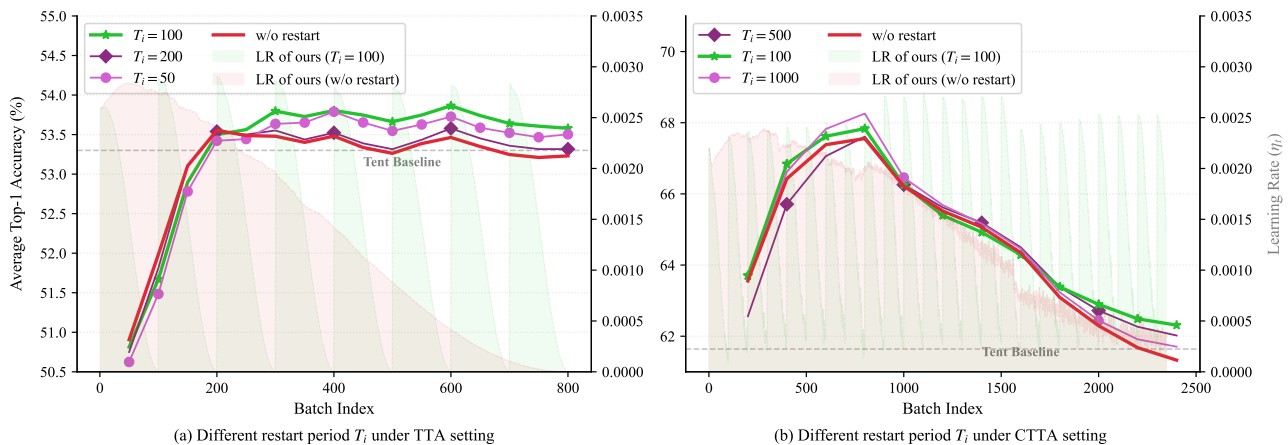

(a) Different restart period $T_i$ under TTA setting        (b) Different restart period $T_i$ under CTTA setting

*Figure 4.* Comparison of different restart periods $T_i$ on ImageNet-Sketch (Standard TTA) and a sequential noise stream (Continual TTA). Shadows at the background represent the LR decided by our CONGA scheduler with different cycle length (red for no restart, green for cycle length 100). The lines represent TTA accuracy under different restart cycles $T_i$, on which the dots represent the restart points. (a) TTA with static target distribution. (b) TTA process through distribution shifts of Contrast, Gaussian Noise, and Defocus Blur.

*Table 6.* Component-wise ablation of CONGA with Tent (ViT on ImageNet-Sketch; ResNet on Gaussian Noise).

|  | Components | | ViT | | ResNet | |
|---|---|---|---|---|---|---|
| Cycle Regulator | Learning Gate | Safety Gate | Acc. | Gain | Acc. | Gain |
|  |  |  | 49.1 | - | 29.9 | - |
| ✓ |  |  | 49.2 | +0.1 | 30.4 | +0.5 |
| ✓ | ✓ |  | 49.5 | +0.4 | 30.5 | +0.6 |
| ✓ | ✓ | ✓ | 50.7 | +1.6 | 34.2 | +4.3 |

*Table 7.* Sensitivity analysis of different base LR $\eta_{base}$ on ImageNet-A (ViT). The highlighted column denotes the selected hyperparameter for our method.

|  | 5e-4 | 8e-4 | 9e-4 | 1e-3 | 2e-3 | 3e-3 |
|---|---|---|---|---|---|---|
| Tent | 51.9 | 52.7 | 52.9 | 52.9 | 53.0 | 54.2 |
| + Ours | 52.7 | 53.7 | 54.0 | 53.6 | 54.5 | 54.5 |
| DeYO | 54.0 | 55.2 | 55.2 | 55.2 | 55.2 | 55.3 |
| + Ours | 54.9 | 55.5 | 55.8 | 55.5 | 55.2 | 55.4 |

performance with different static $\eta_{max}$, revealing that TTA baselines are sensitive to the choice of LR. For instance, while Tent maintains relative stability, DeYO exhibits a sharp performance degradation as the ratio $\eta_{max}/\eta_{base}$ increases beyond 4. Such high sensitivity makes determining $\eta_{max}$ critical in diverse TTA scenarios. Our adaptive CONGA successfully identifies the "sweet spot" for both baselines. By dynamically recalibrating the $\eta_{max}$ based on gradient feedback, our approach automatically lands on the near-optimal ratio, achieving superior accuracy gains without any manual grid search (illustrated by the star markers). In contrast to the upper bound, the sensitivity analysis in Fig. 3 (c) shows that the model is relatively robust to the LR lower bound $\eta_{min}$ across different data quality. The accuracy curves for both high- and low-confidence data remain comparatively flat across a wide range of $\eta_{min}$ ratios. Despite this diminished sensitivity, our adaptive method consistently tracks the empirical optimal $\eta_{min}$, as depicted by the star markers in Fig. 3(c).

**Analysis of different decay strategies.** Tab. 5 evaluates the impact of different functional forms of learning rate decay (introduced in Tab. 1) on TTA performance. The Cosine schedule consistently yields the best results, achieving $50.1\%$ on ImageNet-Sketch and $53.6\%$ on ImageNet-A, out-

performing the constant baseline and other common decay strategies such as Step, Linear and Exponential. Interestingly, we observe that aggressive decay functions such as **Exp** and **Step** can sometimes lead to inferior performance compared to the baseline (e.g., $48.4\%$ vs. $49.1\%$ on Sketch), suggesting that maintaining a sufficient learning rate is crucial for continuous adaptation, especially in the late stages.

**Component-wise ablation.** We also provide a component-wise ablation study in Tab. 6. As shown, each component contributes positively to the final performance: the Cycle Regulator provides a modest but consistent gain, the Learning Gate further improves accuracy, and the Safety Gate delivers the most significant improvement. Importantly, this progressive enhancement is observed across diverse network architectures, which underscores the architecture-agnostic robustness of our design. These results confirm that all three components are necessary and complementary.

### 4.4. Sensitivity Analysis

**Base learning rate.** Tab. 7 evaluates the robustness of our method across different base learning rates (LR). CONGA uniformly improves on both Tent and DeYO baselines across all LR values tested. Moreover, while the performance of Tent ranges from 51.9% to 54.2% with varying

*Table 8.* Sensitivity of $\tau_C$ on ImageNet-Sketch (ViT). The "baseline" column denotes the original method utilizing a standard constant learning rate without CONGA. The highlighted column denotes the selected hyperparameter for our method.

|       | Baseline | 0.95 | 0.9 | 0.85 | 0.8 | 0.75 |
|-------|----------|------|-----|------|-----|------|
| DeYO  | 52.2     | 53.0 | 53.0 | 52.6 | 53.2 | 52.8 |
| Tent  | 49.1     | 50.7 | 50.6 | 50.3 | 50.4 | 49.9 |

*Table 9.* Sensitivity to Safe Guard with DeYO on ImageNet-Sketch (ViT). The highlighted column denotes the selected hyperparameter for our method.

| Setting | $\eta_{max,t}$ Threshold | $\rho_t$ Threshold | Acc. |
|---------|--------------------------|--------------------|------|
| Highly Strict | $1.5 \times \eta_{base}$ | 0.6 | 52.0 |
| Strict | $2 \times \eta_{base}$ | 0.7 | 52.7 |
| Moderate | $3 \times \eta_{base}$ | 0.75 | 53.1 |
| Loose | $4 \times \eta_{base}$ | 0.8 | 53.2 |
| Very Loose | $5 \times \eta_{base}$ | 0.9 | 53.0 |

base LRs (indicating a 2.3% fluctuation), incorporating CONGA narrows this gap to 1.8%. A similar stabilizing effect is observed with DeYO, where our method reduces the accuracy fluctuation from 1.3% to 0.9%. This demonstrates that our scheduling mechanism not only boosts overall accuracy but also reduces the model's sensitivity to initial hyperparameter selection, an essential property for reliable TTA where ground-truth validation data is unavailable.

**Confidence threshold $\tau_C$.** Tab. 8 investigates the impact of the hyperparameter $\tau_C$ on ImageNet-Sketch. Impressively, integrating CONGA consistently outperforms the respective baselines across all tested values of $\tau_C$. For example, when applied to Tent, our method elevates the baseline accuracy (49.1%) to a range of 49.9% to 50.7%. Similarly, with the DeYO baseline (52.2%), CONGA yields robust gains, achieving up to 53.2%. More importantly, performance remains highly stable across the range of $\tau_C \in [0.75, 0.95]$, with minimal accuracy fluctuations.

**Safe Guard.** Tab. 9 demonstrates the robustness of CONGA to Safe Guard strictness, with performance remaining highly stable (52.7% ∼ 53.2%) across a broad range. While the Loose setting yields a marginal peak accuracy (53.2%), we adopt the Moderate level (53.1%) as the default. This conservative choice prioritizes long-term stability against noisy gradients in continuous test streams, mitigating the risk of model collapse without sacrificing competitive accuracy.

### 4.5. Further Analysis

**Restart analysis.** Fig. 4 underscores the critical role of periodic restarts in maintaining model plasticity across different TTA environments. In the standard TTA setting (Fig. 4(a)), although the adaptation target is relatively static, the "w/o

*Table 10.* Comparison of different optimizers with and without our method across various datasets.

| Optimizer | C | R | A | Sketch | Avg. | Δ |
|-----------|------|------|------|--------|------|------|
| SGD | 63.4 | 60.3 | 50.9 | 46.3 | 55.2 | |
| + Ours | 64.0 | 61.1 | 51.3 | 47.1 | 55.9 | +0.7 |
| Adam | 71.2 | 70.0 | 56.9 | 53.3 | 62.9 | |
| + Ours | 71.2 | 70.4 | 57.6 | 53.6 | 63.2 | +0.3 |
| Sign SGD | 71.2 | 69.9 | 56.5 | 53.2 | 62.7 | |
| + Ours | 71.3 | 70.3 | 57.5 | 53.5 | 63.2 | +0.5 |

restart" schedule (red line) tends to settle into suboptimal plateaus. In contrast, our periodic cycling mechanism allows the model to escape local minima. This advantage becomes more pronounced in the CTTA scenario (Fig. 4(b)), where the configuration without restarts suffers from loss of plasticity. Our method periodically resets the optimization trajectory to restore a sufficiently large LR for effective learning in late stages. The results also demonstrate CONGA's performance under various cycle length ($T_i$) choices. The restarts should be frequent enough to capture domain transitions while allowing for sufficient in-cycle convergence.

**Analysis of different optimizers.** Tab. 10 evaluates the compatibility of our method with various mainstream optimizers. Our approach consistently yields positive gains across all benchmarks, regardless of the underlying update rule. When integrated with advanced optimizers like Adam (Kingma & Ba, 2014) and SignSGD (Bernstein et al., 2018), our method further improves the performance by an average of +0.3% and +0.5%, respectively. The most significant enhancement is observed on the challenging ImageNet-A dataset, where our method facilitates more stable convergence under severe distribution shifts. These results underscore that our CONGA acts as a complementary booster to improve existing optimizers for TTA.

## 5. Conclusion

In this work, we revisit the challenge of TTA from the more fundamental perspective of model optimization, revealing the necessity of an adaptive learning rate (LR) scheduler in TTA. Based on our error analysis, we introduce the CONfidence-and-Gradient-Aware learning rate schedule (CONGA), a train-free LR schedule strategy to ensure model plasticity and stability during adaptation. CONGA is composed of two LR gates to ensure effective and safe model updates, then adopts a cosine-decay strategy and periodic restarts to specify the LR for each update. Extensive experiments demonstrate that CONGA serves as a universal, plug-and-play module, consistently enhancing the stability and performance of existing TTA baselines without additional computational overhead. We hope this study inspires further exploration on adaptive optimization for more robust and efficient test-time generalization.

## Acknowledgements

This work was supported in part by the National Natural Science Foundation of China under Grant 62572102, 52441801, in part by the Fundamental Research Funds for the Central Universities (UESTC) under Grant ZYGX2024Z008, in part by Science and Technology Plan Program of Huzhou City 2024GZ06.

## Impact Statement

This paper presents work whose goal is to advance the field of Machine Learning, specifically focusing on the stability and efficiency of Test-Time Adaptation (TTA). There are several potential societal consequences of our work. First, by improving model reliability in dynamic environments through our CONGA schedule, this research contributes to the safety of autonomous systems (e.g., medical diagnostics or smart infrastructure) where distribution shifts often lead to catastrophic failures. Second, our method is highly resource-efficient, maintaining a $1\times$ BP overhead and negligible memory footprint ($\leq 0.15$ MB), which aligns with the "Green AI" initiative by reducing energy consumption during long-term model deployment on edge devices. Ethically, while TTA enhances adaptation, users should remain vigilant about potential "over-adaptation" to biased test-time distributions. While our proposed Learning Gate helps mitigate noise-induced instability, we encourage further monitoring to ensure that adapted models do not inadvertently amplify existing data biases in sensitive social contexts.

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

**Algorithm 1** Pipeline of CONGA

1: **Input:** Test stream $\mathcal{D}_{\text{test}}$ (total $K$ batches), pre-trained model $f_{\theta_0}$, TTA loss $\mathcal{L}_{\text{tta}}$, initial episode length $T$.
2: **Output:** Predictions $\{\hat{y}_j\}_{j=1}^{K}$.
3: Initialize $\theta \leftarrow \theta_0$; initialize EMA statistics; initialize global adaptation step $k \leftarrow 0$ and within-episode step $t \leftarrow 0$.
4: **for** each batch $\mathcal{X}$ in $\mathcal{D}_{\text{test}}$ **do**
5:     **Forward:** compute predictions $p \leftarrow f_\theta(\mathcal{X})$ and loss $\mathcal{L} \leftarrow \mathcal{L}_{\text{tta}}(\theta; \mathcal{X})$; store prediction $\hat{y}_k$.
6:     Compute $\Phi(t)$ via Eq. (2), $\eta_{\max}^k$ via Eq. (3) and $\eta_{\min}^k$ via Eq. (6).
7:     Compute $\eta_k$ via Eq. (7) with phase $t$.
8:     Set optimizer learning rate to $\eta_k$ and take one optimization step on.
9:     $k \leftarrow k+1; t \leftarrow t+1$.
10:    **if** $t = T$ **then**
11:       $t \leftarrow 0$.
12:    **end if**
13: **end for**

# A. Theoretical Definitions and Objective Functions

In this section, we provide the proof of theorem used in our method and detail the objective functions of the baseline TTA methods referenced in the main text.

## A.1. Proof of Theorem 3.1

**Theorem 3.1 (Dynamics of Tracking Error).** *Let $e_t \triangleq \theta_t - w_t$ denote the tracking error at time $t$. Under the assumption of a stochastic update with learning rate $\eta_t$ and a target drift magnitude $\|\delta_t\| = \|w_{t+1} - w_t\| \leq \delta$, suppose the stochastic gradient noise $\kappa_t$ admits a decomposition $\kappa_t = b_t + \xi_t$, where stochastic noise $\xi_t$ satisfies $\mathbb{E}[\xi_t] = 0$, $\mathbb{E}[\|\xi_t\|^2] = \sigma_t^2$, and the bias term is bounded as $\|b_t\| \leq B$. Then, the expectation of the Mean Squared Error at time $t+1$ satisfies the following recurrence relation:*

$$\mathbb{E}[\|e_{t+1}\|^2] \leq (1 - \eta_t)\mathbb{E}[\|e_t\|^2] + \eta_t^2 \sigma_t^2 + \mathcal{O}(\eta_t B^2 + \frac{\delta^2}{\eta_t}). \tag{8}$$

*Proof.* Consider the quadratic local loss $f_t(\theta) = \frac{1}{2}\|\theta - w_t\|^2$ as a local second-order approximation. The parameter update rule is:

$$\theta_{t+1} = \theta_t - \eta_t(\theta_t - w_t + \kappa_t), \tag{9}$$

where $\kappa_t = b_t + \xi_t$. Defining the tracking error $e_t = \theta_t - w_t$ and noting $w_{t+1} = w_t + \delta_t$, the dynamics satisfy:

$$e_{t+1} = (1 - \eta_t)e_t - (\eta_t b_t + \delta_t) - \eta_t \xi_t. \tag{10}$$

Taking the squared norm and expectation on both sides of Eq. 10:

$$\mathbb{E}[\|e_{t+1}\|^2] = (1-\eta_t)^2 \mathbb{E}[\|e_t\|^2] + \mathbb{E}[\|\eta_t b_t + \delta_t\|^2] + \eta_t^2 \mathbb{E}[\|\xi_t\|^2]$$
$$\underbrace{-2(1-\eta_t)\mathbb{E}[\langle e_t, \eta_t b_t + \delta_t \rangle]}_{\text{Cross Term } I} \underbrace{-2(1-\eta_t)\eta_t \mathbb{E}[\langle e_t, \xi_t \rangle] + 2\eta_t \mathbb{E}[\langle \eta_t b_t + \delta_t, \xi_t \rangle]}_{\text{Cross Term } II}. \tag{11}$$

We bound the terms as follows:

1. **Independence of Stochastic Noise:** Since $\xi_t$ is zero-mean and independent of $e_t$, $b_t$, and $\delta_t$, all terms in *Cross Term II* vanish, i.e., $\mathbb{E}[\langle e_t, \xi_t \rangle] = 0$ and $\mathbb{E}[\langle \eta_t b_t + \delta_t, \xi_t \rangle] = 0$.

2. **Young's Inequality for Bias and Drift:** To handle *Cross Term I*, we apply Young's inequality, $-2\langle u, v \rangle \leq \alpha \|u\|^2 + \frac{1}{\alpha}\|v\|^2$, for any $\alpha > 0$. Let $u = (1-\eta_t)e_t$ and $v = \eta_t b_t + \delta_t$:

$$-2\mathbb{E}[\langle (1-\eta_t)e_t, \eta_t b_t + \delta_t \rangle] \leq \alpha(1-\eta_t)^2 \mathbb{E}[\|e_t\|^2] + \frac{1}{\alpha}\mathbb{E}[\|\eta_t b_t + \delta_t\|^2]. \tag{12}$$

3. **Aggregation and Variance Bound:** Substituting the above and the variance bound $\mathbb{E}[\|\xi_t\|^2] = \sigma_t^2$:

$$\mathbb{E}[\|e_{t+1}\|^2] \leq (1+\alpha)(1-\eta_t)^2\mathbb{E}[\|e_t\|^2] + \eta_t^2\sigma_t^2 + (1+\frac{1}{\alpha})\mathbb{E}[\|\eta_t b_t + \delta_t\|^2]. \tag{13}$$

By choosing $\alpha = \eta_t$ (for small $\eta_t$), the contraction coefficient becomes $(1+\eta_t)(1-\eta_t)^2 = (1-\eta_t)(1-\eta_t^2) \leq (1-\eta_t)$. Then

$$\mathbb{E}[\|e_{t+1}\|^2] \leq (1-\eta_t)\mathbb{E}[\|e_t\|^2] + \eta_t^2\sigma_t^2 + (1+\frac{1}{\eta_t})\mathbb{E}[\|\eta_t b_t + \delta_t\|^2]. \tag{14}$$

4. **Drift and Bias Bound:** Using $\|x + y\|^2 \leq 2\|x\|^2 + 2\|y\|^2$, the term $(1+\frac{1}{\eta_t})\mathbb{E}[\|\eta_t b_t + \delta_t\|^2]$ is bounded by $(1+\frac{1}{\eta_t})(2\eta_t^2 B^2 + 2\delta^2)$, which is $\mathcal{O}(\eta_t B^2 + \frac{\delta^2}{\eta_t})$.

Combining these, and noting that for standard TTA settings where $\delta$ and $B$ are sufficiently small, we obtain:

$$\mathbb{E}[\|e_{t+1}\|^2] \leq (1-\eta_t)\mathbb{E}[\|e_t\|^2] + \eta_t^2\sigma_t^2 + \mathcal{O}(\eta_t B^2 + \frac{\delta^2}{\eta_t}). \tag{15}$$

$\square$

## A.2. Proof of Theorem 3.2

**Theorem 3.2 (Optimal Learning Rate for Error Reduction)** *Let $\Delta\mathcal{E}_t \triangleq \mathbb{E}[\|e_t\|^2] - \mathbb{E}[\|e_{t+1}\|^2]$ be the expected error reduction. Under the recurrence relation in Theorem 3.1, the optimal learning rate $\eta_t^*$ that maximizes the lower bound of $\Delta\mathcal{E}_t$ satisfies:*

$$\eta_t^* \propto \frac{1}{\sigma_t^2}, \tag{16}$$

*Proof.* Starting from the recurrence relation established in Theorem 3.1:

$$\mathbb{E}[\|e_{t+1}\|^2] \leq (1-\eta_t)\mathbb{E}[\|e_t\|^2] + \eta_t^2\sigma_t^2 + \lambda\eta_t B^2 + \lambda\frac{\delta^2}{\eta_t}, \tag{17}$$

we define the expected error reduction as $\Delta\mathcal{E}_t \triangleq \mathbb{E}[\|e_t\|^2] - \mathbb{E}[\|e_{t+1}\|^2]$. By rearranging the terms, we obtain a lower bound for the error reduction:

$$
\begin{aligned}
\Delta\mathcal{E}_t &\geq \eta_t\mathbb{E}[\|e_t\|^2] - \eta_t^2\sigma_t^2 - \lambda\eta_t B^2 - \lambda\frac{\delta^2}{\eta_t} \\
&= \eta_t(\mathbb{E}[\|e_t\|^2] - \lambda B^2) - \eta_t^2\sigma_t^2 - \lambda\frac{\delta^2}{\eta_t}.
\end{aligned} \tag{18}
$$

To find the optimal learning rate $\eta_t^*$ that maximizes this lower bound, we define the objective function $g(\eta_t) = \eta_t(\mathbb{E}[\|e_t\|^2] - \lambda B^2) - \eta_t^2\sigma_t^2 - \lambda\frac{\delta^2}{\eta_t}$ and take its first-order derivative with respect to $\eta_t$:

$$\frac{dg(\eta_t)}{d\eta_t} = (\mathbb{E}[\|e_t\|^2] - \lambda B^2) - 2\eta_t\sigma_t^2 + \lambda\frac{\delta^2}{\eta_t^2} = 0. \tag{19}$$

Rearranging the terms, we obtain:

$$(\mathbb{E}[\|e_t\|^2] - \lambda B^2)\eta_t^2 - 2\sigma_t^2\eta_t^3 + \lambda\delta^2 = 0 \tag{20}$$

**Signal-to-Bias Condition.** We assume the *Informativeness Condition* holds, i.e., $\mathbb{E}[\|e_t\|^2] > \lambda B^2$. This is a standard prerequisite for effective adaptation, ensuring that the tracking signal (the distance to the optimum) dominates the systematic bias introduced by the loss function or domain shift. If this condition is violated, the gradient becomes uninformative, and the model inevitably diverges regardless of the step size.

**Operational Regime and Drift.** In the operational regime of continuous test-time adaptation, the target drift $\delta$ (the shift of optimal parameters between steps) is typically infinitesimal compared to the current tracking error. Thus, the term $\lambda\delta^2$ acts

as a vanishingly small perturbation to the stationary point of the objective function. By focusing on the dominant terms in Eq. (20), we arrive at the following approximation for the optimal step size:

$$2\eta_t^* \sigma_t^2 \approx \mathbb{E}[\|e_t\|^2] - \lambda B^2 \implies \eta_t^* \approx \frac{\mathbb{E}[\|e_t\|^2] - \lambda B^2}{2\sigma_t^2}. \tag{21}$$

**Conclusion.** The above relationship indicates that for a given tracking signal and systematic bias, the optimal learning rate is primarily scaled by the inverse of the stochastic gradient noise $\sigma_t^2$:

$$\eta_t^* \propto \frac{1}{\sigma_t^2}. \tag{22}$$

$\square$

## A.3. Baseline Objective Functions

Our proposed CONGA scheduler is designed to be loss-agnostic. In the experiments, we integrated it with the following state-of-the-art TTA objectives:

**Tent (Entropy Minimization) (Wang et al., 2020):** Tent optimizes the model by minimizing the Shannon entropy of the model predictions to increase confidence, while updating only normalization-related parameters at test time.

$$\mathcal{L}_{Tent} = -\frac{1}{B} \sum_{i=1}^{B} \sum_{k=1}^{K} p(y = k|x_i) \log p(y = k|x_i) \tag{23}$$

where $K$ is the number of classes. While effective, entropy minimization is prone to error accumulation when the initial predictions are incorrect.

**DeYO (Pseudo-Label Weighted Entropy) (Lee et al., 2024):** DeYO improves upon Tent by introducing a weighting mechanism based on the consistency of pseudo-labels and minimizing the entropy only for reliable samples.

$$\mathcal{L}_{DeYO} = -\frac{1}{B} \sum_{i=1}^{B} w(x_i) \cdot \sum_{k=1}^{K} p(y = k|x_i) \log p(y = k|x_i) \tag{24}$$

where $w(x_i) \in \{0, 1\}$ denotes a binary reliability indicator derived from DeYO's Pseudo-Label Refinement (PLR) module, which determines whether entropy minimization is applied to a given sample.

**EATA (Sample-Filtered Adaptation with Fisher Regularization) (Niu et al., 2022):** EATA improves adaptation efficiency by filtering out unreliable samples with high entropy and addresses catastrophic forgetting by constraining important parameters. It minimizes the prediction entropy on reliable samples while regularizing the model parameters $\theta$ via a Fisher-information-based weight:

$$\mathcal{L}_{EATA} = -\frac{1}{|\mathcal{X}_{rel}|} \sum_{x \in \mathcal{X}_{rel}} \sum_{k=1}^{K} \hat{y}_k \log \hat{y}_k + \beta \sum_j F_j(\theta_j - \theta_{S,j})^2 \tag{25}$$

where $\mathcal{X}_{rel} = \{x | H(x) < H_0\}$ denotes the set of reliable test samples whose entropy is below a predefined threshold $H_0$. The second term is the **Anti-forgetting Regularizer**, where $F_j$ represents the diagonal Fisher information of the $j$-th parameter $\theta_j$ calculated on the source domain, and $\theta_{S,j}$ denotes the original parameters of the pre-trained source model.

**FOA (Feature Alignment via Statistical Moment Regularization) (Niu et al., 2024):** FOA employs a derivative-free covariance matrix adaptation (CMA) evolution strategy. Its "fitness function" (loss) is a combination of **prediction entropy** and **feature distribution discrepancy**. It minimizes the following objective to perform test-time adaptation via a derivative-free CMA-ES strategy:

$$\mathcal{L}_{FOA} = -\frac{1}{B} \sum_{i=1}^{B} \sum_{k=1}^{K} \hat{y}_{i,k} \log \hat{y}_{i,k} + \lambda \sum_{l=1}^{L} \left( \|\mu_l(\mathcal{X}_t) - \mu_l^S\|_2 + \|\sigma_l(\mathcal{X}_t) - \sigma_l^S\|_2 \right) \tag{26}$$

where $\hat{y}$ is the model prediction, and $\mu_l, \sigma_l$ represent the mean and standard deviation of features at layer $l$, aligning the test batch statistics $\mathcal{X}_t$ with the source statistics $S$.

*Table 11.* Detailed Hyperparameter Settings. "Const." denotes a constant learning rate, and "CONGA" denotes our proposed scheduler. Note that for Segmentation on ACDC, we follow the CoTTA protocol.

| Task | Method | Backbone | Optimizer | Batch Size | Schedule | Base LR ($\eta_{base}$) |
|---|---|---|---|---|---|---|
| TTA | baselines | ResNet-50-BN | SGD | 64 | Constant | $2.5 \times 10^{-4}$ |
| | **Ours** (w/ baselines) | ResNet-50-BN | SGD | 64 | **CONGA** | $2.0 \times 10^{-3}$ |
| | baselines | ViT-Base | SGD | 64 | Constant | $1.0 \times 10^{-3}$ |
| | **Ours** (w/ baselines) | ViT-Base | SGD | 64 | **CONGA** | $1.0 \times 10^{-3}$ |
| CTTA | EATA | ResNet-50-BN | SGD | 64 | Constant | $2.5 \times 10^{-4}$ |
| | **Ours** (w/ EATA) | ResNet-50-BN | SGD | 64 | **CONGA** | $1.0 \times 10^{-3}$ |
| | Tent / Reservoir | ViT-Base | SGD | 64 | Constant | $1.0 \times 10^{-3}$ |
| | **Ours** (w/ Tent/Reservoir) | ViT-Base | SGD | 64 | **CONGA** | $1.0 \times 10^{-3}$ |
| CTTA | PALM | ResNet-50-BN | SGD | 64 | Constant | $5 \times 10^{-5}$ |
| | **PALM** | ViT-Base | SGD | 64 | **CONGA** | $2.5 \times 10^{-4}$ |

**Note:**

- For Reservoir, we use the baseline of Reservoir + EATA.

- For all SGD optimizers in classification and segmentation, we use a momentum of 0.9.

- For the ResNet-50-BN backbone, we set $\eta_{base} = 2.0 \times 10^{-3}$ while the baseline uses a constant LR of $2.5 \times 10^{-4}$. This adjustment accounts for the fact that gradient magnitudes in ResNet are typically larger than those in ViT architectures. By adopting a higher $\eta_{base}$, we ensure that the *mean* learning rate throughout the entire adaptation schedule remains approximately at the $2.5 \times 10^{-4}$ level, facilitating a fair comparison of optimization dynamics.

# B. Detailed Implementation Setup

## B.1. Datasets and Protocols

We evaluate our method on the ImageNet-C benchmark for corruption robustness, and ImageNet-A/R/Sketch for out-of-distribution generalization. Following standard protocols, the adaptation is performed in an online manner with a batch size of 64. The model encounters each test batch only once and updates its parameters without accessing the original source data.

## B.2. Baselines and Hyperparameters

We strictly follow the experimental settings of the baseline methods. For our proposed CONGA scheduler, we only modify the learning rate schedule while keeping the optimizer and other hyperparameters consistent with the baselines for a fair comparison. Tab. 11 details the specific hyperparameter configurations for all experiments.

## B.3. Computational Resources

All experiments were conducted on NVIDIA GeForce RTX 4060 Laptop GPU. The implementation is based on PyTorch. For the efficiency analysis (SignSGD vs. Adam), memory usage was measured using the standard PyTorch profiler.

*Table 12.* Tracking error of toy optimization example (lower the better).

| | Fixed LR | Standard Cosine | CONGA (ours) |
|---|---|---|---|
| pre-shift noisy peak ($t = 20$) | 3.401 | 0.23 | 0.235 |
| post-shift noisy peak ($t = 47$) | 1.776 | 0.646 | 0.477 |
| Average over 80 steps | 0.925 | 0.803 | 0.381 |

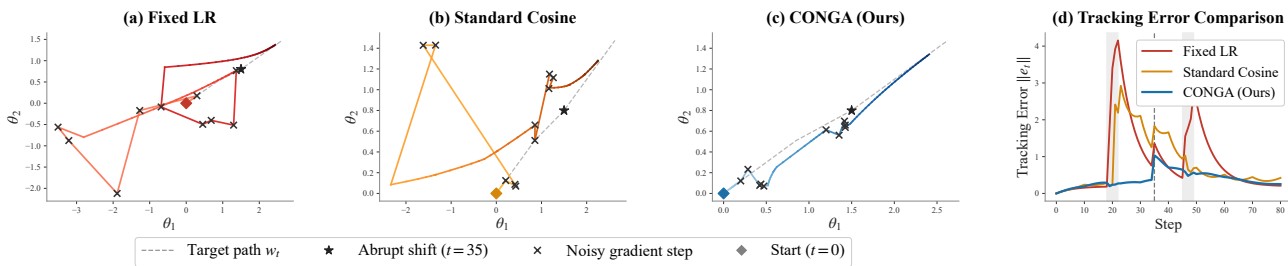

*Figure 5.* Visualizations of the optimization trajectory. The horizontal and vertical axes represent the two model parameters, with color gradients indicating time progression.

*Table 13.* Average accuracy (%) under ImageNet-C (level 5) in the standard TTA setting.

| | Noise | | | Blur | | | | Weather | | | | Digit | | | | Avg | |
|---|---|---|---|---|---|---|---|---|---|---|---|---|---|---|---|---|---|
| Method | Gau. | Shot | Imp. | Def. | Glass | Motion | Zoom | Snow | Frost | Fog | Bright | Contrast | Elastic | Pixel | JPEG | Avg | Δ |
| **ResNet-50-BN base** | | | | | | | | | | | | | | | | | |
| NoAdapt | 2.2±0.1 | 2.9±0.1 | 1.8±0.1 | 17.9±0.0 | 9.8±0.1 | 14.8±0.1 | 22.5±0.1 | 16.9±0.1 | 23.3±0.1 | 24.4±0.1 | 58.9±0.1 | 5.4±0.1 | 16.9±0.1 | 20.7±0.0 | 31.7±0.1 | 18.0±0.1 | |
| SAR | 30.9±0.2 | 33.3±0.0 | 31.5±0.0 | 27.5±0.0 | 27.3±0.0 | 42.2±0.1 | 49.5±0.0 | 47.8±0.1 | 42.5±0.2 | 57.3±0.2 | 67.4±0.2 | 38.1±0.1 | 54.6±0.1 | 58.7±0.1 | 52.3±0.0 | 44.1±0.0 | |
| Tent | 29.9±0.2 | 32.0±0.0 | 31.3±0.0 | 28.0±0.1 | 27.4±0.1 | 41.3±0.1 | 49.2±0.0 | 47.0±0.0 | 41.4±0.0 | 57.7±0.2 | 67.5±0.1 | 28.0±0.0 | 54.8±0.2 | 58.6±0.1 | 52.7±0.2 | 43.2±0.1 | |
| **+ Ours** | 34.2±0.0 | 36.1±0.1 | 35.5±0.0 | 31.1±0.1 | 31.2±0.0 | 46.0±0.1 | 51.7±0.2 | 50.9±0.2 | 43.2±0.0 | 59.4±0.0 | 68.0±0.1 | 27.9±0.0 | 57.6±0.0 | 60.6±0.1 | 55.0±0.0 | 45.9±0.1 | +2.7 |
| EATA | 35.6±0.2 | 37.4±0.3 | 36.6±0.1 | 33.8±0.5 | 33.1±0.4 | 47.0±0.4 | 52.7±0.0 | 51.4±0.1 | 45.5±0.0 | 59.9±0.0 | 68.2±0.0 | 44.3±0.1 | 58.0±0.2 | 60.5±0.3 | 55.2±0.1 | 47.9±0.2 | |
| **+ Ours** | 35.6±0.0 | 38.2±0.0 | 36.9±0.1 | 33.5±0.2 | 33.7±0.0 | 47.7±0.1 | 52.8±0.1 | 52.2±0.3 | 45.7±0.0 | 60.3±0.2 | 67.8±0.0 | 44.5±0.1 | 58.3±0.0 | 60.7±0.3 | 55.2±0.1 | 48.2±0.1 | +0.3 |
| Deyo | 35.5±0.2 | **38.6±0.1** | **38.1±0.1** | **33.3±0.2** | 32.8±0.2 | **48.5±0.1** | 52.9±0.1 | 52.4±0.0 | 46.2±0.1 | 60.5±0.0 | **68.1±0.1** | 41.5±0.1 | 58.4±0.1 | **61.4±0.1** | 55.7±0.1 | 48.3±0.0 | |
| **+ Ours** | **36.1±0.1** | 38.5±0.0 | 37.5±0.1 | 33.2±0.0 | **33.0±0.0** | **48.5±0.0** | 52.9±0.2 | **52.8±0.0** | 46.3±0.0 | 60.6±0.1 | 67.5±0.1 | **45.8±0.0** | **58.6±0.1** | 61.3±0.1 | **55.9±0.0** | **48.6±0.1** | +0.3 |
| **ViT base** | | | | | | | | | | | | | | | | | |
| NoAdapt | 56.8±0.0 | 56.8±0.0 | 57.5±0.0 | 46.9±0.0 | 35.6±0.0 | 53.1±0.0 | 44.8±0.0 | 62.2±0.0 | 62.5±0.0 | 65.7±0.0 | 77.7±0.0 | 32.6±0.0 | 46.0±0.0 | 67.0±0.0 | 67.6±0.0 | 55.5±0.0 | |
| LAME | 56.5±0.0 | 56.6±0.0 | 57.3±0.4 | 46.±0.34 | 34.8±0.1 | 52.7±0.2 | 44.3±0.1 | 58.4±0.1 | 61.6±0.0 | 63.1±0.0 | 77.5±0.2 | 24.7±0.1 | 44.6±0.0 | 66.6±0.5 | 67.3±0.1 | 54.2±0.0 | |
| T3A | 56.4±0.1 | 56.9±0.2 | 57.3±0.1 | 47.8±0.0 | 37.7±0.5 | 54.2±0.0 | 46.9±0.1 | 63.6±0.2 | 60.8±0.9 | 68.5±0.3 | 78.1±0.0 | 38.4±0.1 | 50.0±0.1 | 67.6±0.2 | 69.0±0.1 | 56.9±0.3 | |
| SAR | 59.2±0.0 | 60.5±0.0 | 60.7±0.0 | 57.5±0.4 | 55.6±0.0 | 61.8±0.2 | 57.6±0.0 | 65.9±0.3 | 63.5±0.1 | 69.1±0.1 | 78.7±0.0 | 45.7±0.2 | 62.4±0.2 | 71.9±0.0 | 70.3±0.0 | 62.7±0.1 | |
| Tent | 60.3±0.1 | 61.6±0.1 | 61.8±0.1 | 59.2±0.1 | 56.5±0.5 | 63.5±0.0 | 59.2±0.1 | 61.5±0.3 | 64.5±0.1 | 16.6±31.5 | 79.3±0.1 | 67.5±0.0 | 61.3±0.3 | 72.6±0.1 | 70.7±0.0 | 60.1±0.1 | |
| **+ Ours** | 61.1±0.1 | 62.7±0.0 | 62.8±0.1 | 60.3±0.2 | 59.0±0.1 | 65.0±0.1 | 61.5±0.1 | 62.9±0.0 | 64.1±0.2 | 46.6±38.6 | 79.8±0.2 | 68.7±0.0 | 65.2±0.1 | 74.2±0.1 | 72.1±0.0 | 64.3±0.1 | +3.2 |
| FOA | 60.9±0.0 | 61.6±0.0 | 62.7±0.1 | 56.5±0.1 | 49.8±0.0 | 60.8±0.1 | 56.4±0.2 | 66.6±0.0 | 63.3±0.0 | 69.2±0.0 | 79.4±0.1 | 64.3±0.0 | 58.2±0.0 | 71.4±0.1 | 70.3±0.1 | 63.4±0.1 | |
| **+ Ours** | 61.2±0.0 | 62.0±0.0 | 62.9±0.0 | 56.8±0.2 | 51.0±0.0 | 61.4±0.0 | 57.2±0.1 | 67.1±0.1 | 63.8±0.0 | 69.8±0.0 | 79.7±0.1 | 64.6±0.0 | 59.5±0.0 | 72.1±0.1 | 70.9±0.1 | 64.0±0.1 | +0.6 |
| Deyo | 62.7±0.5 | 64.1±0.2 | 63.8±0.3 | 60.2±0.0 | 60.7±0.1 | 66.5±0.2 | 62.9±0.1 | 70.9±0.0 | 69.6±0.1 | 73.1±0.2 | **80.6±0.1** | 38.3±26.1 | 69.6±0.1 | 75.7±0.1 | 73.7±0.2 | 66.2±1.9 | |
| **+ Ours** | **62.9±0.3** | **64.3±0.1** | **64.1±0.1** | **60.7±0.2** | **62.2±0.5** | **67.1±0.3** | **64.3±0.1** | **71.9±0.2** | **70.4±0.2** | **73.2±0.1** | **80.6±0.0** | **62.9±2.4** | **72.1±0.4** | **76.6±0.2** | **74.4±0.1** | **68.5±0.3** | +2.3 |

## C. Toy Model

We construct a 2D toy example where a parameter $\theta \in \mathbb{R}^2$ tracks a moving target $w_t$ under a quadratic loss $f_t(\theta) = \frac{1}{2}|\theta - w_t|^2$, mirroring the assumptions of Theorem 3.1. Parameters are updated via gradient descent $\theta_{t+1} = \theta_t - \eta_t \cdot g_t$ with base learning rate $\eta = 0.15$ over $T = 80$ steps. Distribution shift is simulated by an abrupt jump in the target trajectory at $t = 35$: $w_t = [0.025t, 0.015t]^\top$ for $t < 35$, and $w_t = [1.5, 0.8]^\top + [0.025(t - 35), 0.015(t - 35)]^\top$ for $t \geq 35$. Gradient noise is injected at $t \in [18, 22] \cup [45, 49]$ via $g_t = (\theta_t - w_t) + \epsilon_t$ where $\epsilon_t \sim \mathcal{N}(0, 6.0^2)$, simulating unreliable pseudo-labels. Confidence is simulated as $\bar{C}_t = 0.3$ during noisy steps and $\bar{C}_t = 0.88 + 0.08\sin(0.4t)$ otherwise, reflecting the natural inverse relationship between prediction confidence and gradient noise. The EMA momentum is set to 0.85 and the cosine cycle length to $T_i = 10$.

As shown in Tab. 12, at the pre-shift peak ($t = 20$), CONGA and Standard Cosine yield comparable tracking errors, both outperforming Fixed LR. This validates the effectiveness of our Cycle Regulator. At the post-shift noisy peak ($t = 47$), CONGA achieves the lowest tracking error, demonstrating more robust adaptation under concurrent gradient noise and distribution shift. Overall, CONGA achieves the lowest mean tracking error across all 80 steps, confirming its superiority.

To better illustrate the experiment, we provide visualizations of the optimization trajectory in Fig. 5. Fixed LR is severely disrupted by noisy gradients while Standard Cosine Decay struggles to recover after the shift. CONGA maintains a trajectory closely following $w_t$, demonstrating its superiority in handling both gradient noise and distribution shift.

*Table 14.* Per-visit average Top-1 accuracy (%) on ImageNet-C (level 5) during 10 recurring adaptation cycles. "Avg." denotes the mean performance across all visits.

|  | 1 | 2 | 3 | 4 | 5 | 6 | 7 | 8 | 9 | 10 | Avg. |
|---|---|---|---|---|---|---|---|---|---|---|---|
| ResNet |  |  |  |  |  |  |  |  |  |  |  |
| NoAdapt | 31.6 | 31.4 | 31.3 | 31.4 | 31.4 | 31.4 | 31.4 | 31.4 | 31.3 | 31.3 | 31.4 |
| PALM | 37.2 | 41.5 | 42.5 | 42.7 | 42.6 | 42.0 | 41.6 | 41.4 | 40.6 | 39.8 | 41.2 |
| EATA | 41.2 | 41.7 | 42.0 | 41.9 | 42.2 | 41.9 | 41.9 | 42.0 | 42.0 | 41.8 | 41.9 |
| **+Ours** | 41.5 | 42.0 | 42.3 | 42.2 | 42.3 | 42.1 | 42.2 | 42.1 | 42.2 | 42.2 | 42.1 |
| Vit |  |  |  |  |  |  |  |  |  |  |  |
| NoAdapt | 51.3 | 51.5 | 51.5 | 51.4 | 51.4 | 51.4 | 51.4 | 51.3 | 51.4 | 51.3 | 51.4 |
| PALM | 61.0 | 61.9 | 62.8 | 63.7 | 63.9 | 64.1 | 64.2 | 64.4 | 64.4 | 64.4 | 63.5 |
| Tent | 57.7 | 61.1 | 61.7 | 62.0 | 62.3 | 62.3 | 62.3 | 62.4 | 62.4 | 62.4 | 61.7 |
| **+Ours** | 57.9 | 61.4 | 61.9 | 62.2 | 62.4 | 62.4 | 62.5 | 62.5 | 62.6 | 62.6 | 61.8 |
| Reservoir | 60.2 | 62.8 | 63.9 | 64.4 | 64.9 | 62.5 | 65.5 | 65.7 | 65.8 | 66.1 | 64.2 |
| **+Ours** | 61.3 | 63.9 | 64.7 | 65.5 | 65.7 | 66.0 | 66.5 | 66.5 | 66.5 | 66.6 | 65.3 |

*Table 15.* Average classification accuracy (%) under different domains.

| Method | A | $\Delta$ | Sketch | $\Delta$ | R | $\Delta$ | Avg | $\Delta$ |
|---|---|---|---|---|---|---|---|---|
| NoAdapt | $0.1_{\pm0.0}$ |  | $44.9_{\pm0.0}$ |  | $59.5_{\pm0.0}$ |  | $34.8_{\pm0.0}$ |  |
| FOA | $50.9_{\pm0.1}$ |  | $46.3_{\pm0.0}$ |  | $60.3_{\pm0.1}$ |  | $52.6_{\pm0.0}$ |  |
| **+ Ours** | $51.3_{\pm0.1}$ | $+0.4_{\pm0.05}$ | $47.1_{\pm0.1}$ | $+0.8_{\pm0.03}$ | $61.1_{\pm0.2}$ | $+0.8_{\pm0.17}$ | $53.2_{\pm0.1}$ | $+0.6_{\pm0.12}$ |
| TENT | $52.9_{\pm0.1}$ |  | $49.1_{\pm0.2}$ |  | $63.9_{\pm0.0}$ |  | $55.3_{\pm0.1}$ |  |
| **+ Ours** | $53.6_{\pm0.0}$ | $+0.7_{\pm0.04}$ | $50.7_{\pm0.0}$ | $+1.6_{\pm0.11}$ | $65.2_{\pm0.1}$ | $+1.3_{\pm0.05}$ | $56.3_{\pm0.0}$ | $+1.2_{\pm0.08}$ |
| Deyo | $55.2_{\pm0.0}$ |  | $52.2_{\pm0.1}$ |  | $66.1_{\pm0.2}$ |  | $57.8_{\pm0.1}$ |  |
| **+ Ours** | $55.5_{\pm0.0}$ | $+0.3_{\pm0.03}$ | $53.0_{\pm0.1}$ | $+0.8_{\pm0.10}$ | $68.3_{\pm0.0}$ | $+2.2_{\pm0.19}$ | $58.9_{\pm0.0}$ | $+1.1_{\pm0.21}$ |

# D. Additional Results

## D.1. Full Results

We provide the full results, including standard deviations, in Tab. 13 and 15 to complement the summarized data in Tab. 2 and 4. Furthermore, the per-visit average accuracies are detailed in Tab. 14 as a supplement to Tab. 3. Collectively, these comprehensive results validate the effectiveness and robustness of our proposed CONGA.

*Table 16.* Accuracy on ImageNet-C under CTTA and CTTA+CDC scenarios (ViT). Columns represent recurring domain visits.

|  |  | 1 | 2 | 3 | 4 | 5 | 6 | 7 | 8 | 9 | 10 | Avg |
|---|---|---|---|---|---|---|---|---|---|---|---|---|
| | Reservoir | 60.18 | 62.8 | 63.91 | 64.42 | 64.86 | 65.21 | 65.52 | 65.69 | 65.81 | 66.05 | 64.4 |
| CTTA | w/ CONGA | 60.99 | 63.95 | 64.73 | 65.29 | 65.67 | 66.03 | 66.23 | 66.35 | 66.59 | 66.67 | 65.3 |
| | w/ Per-domain CONGA | 61.29 | 63.92 | 64.67 | 65.49 | 65.67 | 66.01 | 66.45 | 66.48 | 66.54 | 66.60 | 65.3 |
| | Reservoir | 56.41 | 58.96 | 59.82 | 60.41 | 61.05 | 61.03 | 61.47 | 61.71 | 61.92 | 62.19 | 60.5 |
| CTTA+CDC | w/ CONGA | 57.01 | 59.52 | 63.51 | 64.65 | 65.19 | 65.52 | 65.65 | 66.00 | 66.23 | 66.24 | 64.0 |
| | w/ Per-domain CONGA | 57.34 | 62.36 | 63.76 | 64.56 | 65.09 | 65.41 | 65.55 | 65.93 | 66.13 | 66.21 | 64.2 |

## D.2. Analysis of dynamic restarts

To investigate the effects of dynamic and our static restarts, we evaluated two variants of CONGA integrated with Reservoir (a domain-detection method for CTTA) across two realistic CTTA scenarios.

- **Evaluated CONGA variants**: CONGA adopts a fixed restart length of 100 regardless of detected domain changes. Per-domain CONGA maintains an LR schedule for each domain detected by Reservoir. The LR schedule for different domains evolves and restarts independently, thus eliminating the influence of domain shifts during restarts. Therefore, Per-domain CONGA serves as a natural example to demonstrate the effectiveness of dynamic restarts.

*Table 17.* Accuracy with batch size=1 on Tent (ViT) and EATA (ResNet).

|        | Pixelate | Elastic | Contrast | Avg. |
|--------|----------|---------|----------|------|
| EATA   | 46.5     | 27.0    | 44.5     | 39.3 |
| +Ours  | 51.2     | 27.9    | 47.1     | 42.1 |
| Tent   | 67.0     | 54.0    | 65.0     | 62.0 |
| +Ours  | 73.6     | 62.6    | 67.7     | 68.0 |

*Table 18.* Efficiency Comparison with other baselines. **BP** denotes the number of backpropagation passes per iteration.

| Setting | Method | Base Loss | Backbone | Optimizer | C | A | R | Sketch | Avg. Acc (↑) | Memory (↓) | Complexity |
|---------|--------|-----------|----------|-----------|------|------|------|--------|--------------|------------|------------|
| **TTA** | Tent | Entropy | ViT | SGD | 61.1 | 52.9 | 63.9 | 49.1 | 56.8 | 0MB | 1× BP |
|         | + SAR |          |     | SAM | 62.7 | 52.5 | 63.3 | 48.7 | 56.8 | 0.29MB | 2× BP |
|         | + Ours |          |     | SGD | 64.3 | 53.6 | 65.2 | 50.7 | 58.5 | 0.15MB | 1× BP |
|         | FOA | Entropy+ | ViT | Sign SGD | 71.2 | 56.5 | 69.9 | 53.2 | 62.7 | 0MB | 1× BP |
|         | + MGG | Regularization |  | Meta-Grad | 71.4 | 56.7 | 70.2 | 53.3 | 62.9 | 22.51MB | 1× BP |
|         | + Ours |          |     | Sign SGD | 71.3 | 57.5 | 70.3 | 53.5 | 63.2 | 0MB | 1× BP |
| **CTTA** | EATA | Sample-Filter | ResNet | SGD | 41.9 | - | - | - | 41.9 | 0MB | 1× BP |
|          | + PALM | Entropy |       | Adam | 41.2 | - | - | - | 41.2 | 4566.82MB | 2× BP |
|          | + Ours |          |       | SGD | 42.1 | - | - | - | 42.1 | 0.2MB | 1× BP |

- **Evaluated CTTA scenarios**: CTTA is the setting where domains change smoothly. CTTA+CDC adopts the Continual Dynamic Change (CDC) setting from Reservoir, bringing abrupt and unpredictable distribution shifts.

As shown in Tab. 16, the two variants achieve comparable overall results across both scenarios. Although global CONGA falls behind in initial stages of CTTA+CDC due to EMA warm-up, the final results remain stable because the periodic high-LR restarts in CONGA's cosine schedule provide sufficient adaptation capacity to recover from the initial misalignment. This mechanism bridges the performance gap introduced by the EMA warm-up within a few visits.

### D.3. Results under more challenging scenarios

**Under challenging CDC scenarios.** As shown in the bottom half of Tab. 16, the baseline Reservoir struggles in this highly dynamic environment, yielding an average accuracy of only $60.5\%$. In stark contrast, integrating CONGA significantly fortifies the model, achieving a superior average accuracy of $64.0\%$ (a notable $+3.5\%$ absolute improvement). More importantly, tracking the performance across recurring domain visits reveals that while the baseline adapts sluggishly (remaining below $60\%$ in early rounds), CONGA dramatically accelerates the adaptation process. It rapidly surges to $63.51\%$ by the third visit and maintains a substantial lead up to the final round ($66.24\%$). This clearly demonstrates that our scheduling mechanism effectively mitigates the adverse impacts of sudden domain shifts, ensuring rapid recovery and stable learning even in extremely challenging environments.

**Under batch size 1.** Tab. 17 evaluates the adaptation performance in the setting where the batch size is limited to 1. In this challenging scenario, TTA methods typically suffer from instability due to the high variance of single-sample gradient estimates and unreliable batch statistics. Nevertheless, our proposed method consistently yields improvements across various corruptions and baselines. For instance, it enhances EATA's average accuracy from $39.3\%$ to $42.1\%$. The gain is even more pronounced when applied to Tent, where our approach achieves a $6.0\%$ absolute improvement (from $62.0\%$ to $68.0\%$). These findings underscore that by dynamically tracking the optimization landscape, our strategy effectively mitigates the noise inherent in single-sample updates, providing robust and superior performance.

### D.4. Efficiency Analysis

As summarized in Table 18, our proposed schedule demonstrates superior computational and memory efficiency on the ResNet-50-BN backbone. Specifically, our method maintains a standard $1\times$ backpropagation (BP) overhead, whereas the stability-oriented baseline **SAR** requires $2\times$ BP yet yields lower accuracy ($58.5\%$ vs. $56.8\%$ when combined with Tent). Moreover, while **MGG** introduces a substantial memory footprint of $22.51$ MB for meta-gradient computation, our approach

incurs negligible memory overhead ($\leq 0.15$ MB). These results underscore that by dynamically tracking the theoretical optimal learning rate, our method significantly enhances adaptation performance (e.g., reaching $63.2\%$ with FOA) without compromising real-time efficiency or memory constraints essential for practical test-time adaptation.

## E. Limitations

While CONGA demonstrates robust and stable performance in various test-time adaptation scenarios, we identify two primary limitations. First, the behavior of CONGA under extremely limited test samples (e.g., fewer than $T_i$) remains an open question. In such data-scarce regimes, the cosine learning rate schedule may not complete a full cycle, and the EMA-based bounds may lack sufficient iterations to properly warm up and stabilize. Second, as a method explicitly designed to recalibrate optimization dynamics, CONGA inherently requires gradient computation during inference. Consequently, it is incompatible with lightweight forward-pass-only TTA approaches that strictly prohibit backpropagation to meet extreme computational or memory constraints on edge devices.

