# OpenReview forum: "CONGA:Confidence-and-Gradient-Aware Learning Rate Schedule for Test Time Adaptation"
_ICML.cc/2026/Conference — ICML 2026 regular_

### Official Review · Reviewer_pzj5 · 2026-02-26

**Soundness:** 2
**Presentation:** 2
**Significance:** 3
**Originality:** 2
**Overall Recommendation:** 4
**Confidence:** 3

**Summary:**

This work proposes a new learning rate scheduler aiming for test-time-adaptation. Besides the cycle regulator, it estimates high and low bounds to smooth the optimization process, where the lower bound is based on confidence and the higher bound is to mitigate overconfidence. There are theories corresponding to the design motivation and experiments showing the effectiveness.

**Compliance With Llm Reviewing Policy:**

Affirmed.

**Final Justification:**

The rebuttal substantially addressed my main concerns, especially by adding component-wise ablations and reporting run-to-run mean±std for the reported gains; while some related-work and positioning issues remain, I now believe the paper’s technical contribution is solid enough for a weak accept.

**Key Questions For Authors:**

What are the limitations of the paper?

Is there a concrete toy optimization example showing how CONGA is superior?

Could the authors clarify how the various fixed thresholds (e.g., confidence cutoff of 0.9, Safe Guard triggers, Ti = 100) and the setting-specific ηbase values were selected? Was a separate validation set used, or were these chosen based on the test benchmarks directly?

**Limitations:**

No. The paper includes a brief Impact Statement but does not discuss technical limitations. A few worth addressing: the Learning Gate relies on batch-level confidence estimates, which may be unreliable under small batch sizes or severe corruption; the cycle length is set heuristically and its sensitivity is not analyzed; and it is unclear how the method behaves when the test stream contains only a handful of samples. Adding a short limitations paragraph discussing these points would strengthen the paper.

**Strengths And Weaknesses:**

Strengths:

There is intuitive motivation for the design of the scheduler with low computation overhead, which can be widely used in transfer learning if proved useful.

The proposed method is tested on several different representative benchmarks.

Weaknesses:

There are no ablation studies for the three components described in Figure 1. This harms soundness.

The paper never specifies how many independent runs were used to compute the reported standard deviations, making it impossible to assess the reliability. More importantly, many improvements attributed to CONGA are marginal (+0.3%, +0.6%), yet no statistical significance tests are conducted to rule out random variation. The main results tables also omit standard deviations entirely, deferring them to the appendix without justification. Without proper significance testing, it remains unclear whether the reported gains reflect a consistent, meaningful benefit. This harms soundness and significance.

The related work overlooks several directly relevant contributions. First, Smith (2017) introduced cyclical learning rates with warm restarts, which is the direct precursor to CONGA's cosine decay and periodic restart design; without engaging this work, it is unclear what is genuinely novel in CONGA's scheduling strategy. Second, the TTT line of work (Sun et al., 2020), a major and well-established TTA paradigm, is entirely absent, leaving the paper's positioning within the broader TTA landscape incomplete. Third, Schneider et al. (2020) pioneered online BN statistics adaptation on ImageNet-C, the primary benchmark used in this paper, and should be discussed as a foundational context. Fourth, RoTTA (Yuan et al., 2023) targets the same stability-plasticity trade-off in continual TTA and represents a directly competing approach that warrants explicit comparison or discussion. This harms presentation and puts originality into question.

Minor: Most notably, the baseline name "Reservoir" is misspelled as "Reservior" in four places, including a table row label and an appendix sentence where both spellings appear within the same line. In Section 3.1, "characterized by the competing between the Error Contraction and the Stochastic Variance" should read "the competition between"; "a error-dominant regime" should be "an error-dominant regime." "LARSoptimizer" (§3.2) should be written as "LARS optimizer." These errors suggest the manuscript needs a careful proofread before publication.

References

Smith, L. N. (2017). Cyclical learning rates for training neural networks. WACV.

Sun, Y., Wang, X., Liu, Z., Miller, J., Efros, A. A., & Hardt, M. (2020). Test-time training with self-supervision for generalization under distribution shifts. ICML.

Schneider, S., Rusak, E., Eck, L., Bringmann, O., Brendel, W., & Bethge, M. (2020). Improving robustness against common corruptions by covariate shift adaptation. NeurIPS.

Yuan, L., Xie, B., & Li, S. (2023). Robust test-time adaptation in dynamic scenarios. CVPR.

---

> ### Author Rebuttal · Authors · 2026-03-31
>
> **Response to Weaknesses:** Thanks for your detailed comments and instructions. We address them as follows:
> - Component-wise ablation studies: We have conducted ablation study in Table 9 in our response to Reviewer xks2. The results show that each component contributes positively to the final performance, which confirms the effectiveness of all three components.
> - Number of independent runs: We use 5 independent runs in all experiments to compute the standard deviation, which we will clarify in the revision.
> - Related work: Thank you for the suggestion. We will add discussions of all four mentioned works in the related work section to fulfill the work.
> - Typos: We will also carefully proofread the manuscript to correct all typos and grammatical errors noted above. Thanks again for your effort.
>
> **Response to Question 1 (limitation)**: Thanks for the question. We first respond to the limitations raised in the review: (1) batch-level confidence reliability: as shown in Table 1 and Table 4 of our response to Reviewer G1Bz, CONGA maintains consistent improvements under both severe corruption and batch size 1. (2) cycle length sensitivity: Fig. 4 in our paper demonstrates that CONGA remains robust across various cycle lengths $T_i$. (3) behavior under limited test samples: we acknowledge that the behavior of CONGA under extremely limited test samples (e.g., fewer than $T_i​$) remains an open question. In such cases, the cosine schedule may not complete a full cycle and the EMA-based bounds may not stabilize. We will discuss this in the revision.
> Beyond the points raised, we identify that CONGA requires gradient computation, making it incompatible with forward-pass-only TTA methods.
>
> **Response to Question 2 (toy example):**
> Thanks for the suggestion. We construct a 2D toy example where a parameter $\theta \in \mathbb{R}^2$ tracks a moving target $w_t$​ under a quadratic loss $f_t(\theta) = \frac{1}{2}\|\theta - w_t\|^2$, mirroring the assumptions of Theorem 3.1. Parameters are updated via gradient descent $\theta_{t+1} = \theta_t - \eta_t \cdot g_t$​ with base learning rate $\eta = 0.15$ over $T=80$ steps. Distribution shift is simulated by an abrupt jump in the target trajectory at $t=35$: $w_t = [0.025t,\ 0.015t]^\top$ for $t < 35$, and $w_t = [1.5, 0.8]^\top + [0.025(t-35),\ 0.015(t-35)]^\top$ for $t \geq 35$. Gradient noise is injected at $t \in [18,22] \cup [45,49]$ via $g_t = (\theta_t - w_t) + \epsilon_t$ where $\epsilon_t \sim \mathcal{N}(0, 6.0^2)$, simulating unreliable pseudo-labels. Confidence is simulated as $C_t = 0.3$ during noisy steps and $C_t = 0.88 + 0.08\sin(0.4t)$ otherwise, reflecting the natural inverse relationship between prediction confidence and gradient noise. The EMA momentum is set to 0.85 and the cosine cycle length to $T_i = 10$.
> As shown in Table 10, at the pre-shift noisy peak (t=20), CONGA and Standard Cosine yield comparable tracking errors, both outperforming Fixed LR. This validates the effectiveness of our Cycle Regulator. At the post-shift noisy peak (t=47), CONGA achieves the lowest tracking error, demonstrating more robust adaptation under concurrent gradient noise and distribution shift. Overall, CONGA achieves the lowest mean tracking error across all 80 steps, confirming its superiority.
> To better illustrate the experiment, we provide [visualizations](https://anonymous.4open.science/r/CONGA-Optimization-toy-model-46A7/conga_toy_paper.png) of the optimization trajectory. The horizontal and vertical axes represent the two model parameters, and the solid lines show the update trajectories with color gradient indicating time progression. Fixed LR is severely disrupted by noisy gradients while Standard Cosine Decay struggles to recover after the shift. CONGA maintains a trajectory closely following $w_t​$, demonstrating its superiority in handling both gradient noise and distribution shift. We will include this example in the revision.
>
> **Table 10: Tracking error of toy optimization example (lower the better).**
> ||Fixed LR|Standard Cosine|CONGA(ours)|
> |-|-|-|-|
> |pre-shift noisy peak (t=20)|3.401|0.23|0.235|
> |post-shift noisy peak (t=47)|1.776|0.646|0.477|
> |Average over 80 steps|0.925|0.803|0.381|
>
> **Response to Question 3 (hyperparameter selection)**:
> Thank you for this question. We clarify the selection of each hyperparameter as follows.
> $\eta_{base}​$: For ViT, we follow the convention of $\eta=10^{-3}$ for TTA (consistent with prior works such as DeYO). For ResNet, we adjusted $\eta_{base}$ due to its larger gradient magnitudes.
> $T_i$, Confidence cutoff and Safety Guard triggers: These were determined on a small held-out subset manually split from ImageNet-R and kept fixed across all datasets and architectures. As shown in Table 3 (confidence cutoff), Table 5 (Safety Guard), and Fig. 4 in our paper ($T_i​$), CONGA is robust to these choices, confirming that our selection is both principled and generalizable across diverse settings.

---

> > ### Author Rebuttal · Reviewer_pzj5 · 2026-04-02
> >
> > Thanks for the clarification. The added component-wise ablation and the note that all experiments use 5 runs are helpful. However, my main concern about reliability is still not fully addressed: many of the gains are quite small (+0.3%, +0.6%), but no statistical significance analysis is provided. Could the authors clarify whether these improvements are statistically significant across the 5 runs, or at least report the corresponding mean ± std more clearly in the main tables?

---

> > > ### Author Response · Authors · 2026-04-02
> > >
> > > **Table 11: Mean and standard deviation of CONGA gains (Δ) on ViT. We report results under both TTA and CTTA settings on ImageNet-C, and TTA results on other datasets. Bold indicates gains larger than 0.5%.**
> > >
> > > ||ImageNet-C (TTA)||ImageNet-C (CTTA)||ImageNet-A||ImageNet-Sketch||ImageNet-R||
> > > |-|:-:|:-:|:-:|:-:|-|-|-|-|-|-|
> > > ||Acc.|Δ|Acc.|Δ|Acc.|Δ|Acc.|Δ|Acc.|Δ|
> > > |Tent|61.1±1.23|-|62.4±0.01|-|52.9±0.05|-|49.1±0.20|-|63.9±0.03|-|
> > > |+Ours|64.3±1.31|**3.2±1.09**|62.6±0.00|0.2±0.00|53.6±0.02|**0.7±0.04**|50.7±0.03|**1.6±0.11**|65.2±0.13|**1.3±0.05**|
> > > |FOA|63.4±0.07|-|-|-|50.9±0.07|-|46.3±0.01|-|60.3±0.09|-|
> > > |+Ours|64.0±0.09|**0.6±0.09**|-|-|51.3±0.05|0.4±0.05|47.1±0.05|**0.8±0.03**|61.1±0.21|**0.8±0.17**|
> > > |DeYO|66.2±1.85|-|-|-|55.2±0.03|-|52.2±0.06|-|66.1±0.23|-|
> > > |+Ours|68.5±0.26|**2.3±1.03**|-|-|55.5±0.04|0.3±0.03|53.0±0.13|**0.8±0.1**|68.3±0.02|**2.2±0.19**|
> > > |Reservoir|-|-|66.0±0.05|-|-|-|-|-|-|-|
> > > |+Ours|-|-|66.6±0.02|**0.6±0.03**|-|-|-|-|-|-|
> > >
> > > **Table 12: Mean and standard deviation of CONGA gains (Δ) on ImageNet-C under TTA and CTTA (ResNet). Bold indicates gains larger than 0.5%.**
> > > ||TTA||CTTA||
> > > |-|:-:|:-:|:-:|:-:|
> > > ||Acc.|Δ|Acc.|Δ|
> > > |Tent|43.2±0.08|-|-|-|
> > > |+Ours|45.9±0.05|**2.7±0.07**|-|-|
> > > |EATA|47.9±0.15|-|41.8±0.06|-|
> > > |+Ours|48.2±0.07|0.3±0.13|42.2±0.03|0.4±0.06|
> > > |DeYO|48.3±0.02|-|-|-|
> > > |+Ours|48.6±0.05|0.3±0.05|-|-|
> > >
> > > We report the average accuracy for each dataset and the improvements ($\Delta$) achieved by integrating our CONGA with mean±std in Table 11 and Table 12 above. The results show that even the smallest gains are statistically reliable: for example, DeYO+Ours achieves +0.3±0.05 on ImageNet-C with ResNet and FOA+Ours achieves +0.6±0.09 on ImageNet-C with ViT, where the std of $\Delta$ is far smaller than the mean improvement, indicating consistent gains across all 5 runs. Furthermore, the majority of improvements are substantially larger (e.g., +2.7±0.07 for Tent on ImageNet-C with ResNet), demonstrating that CONGA provides both statistically reliable and practically meaningful gains across diverse settings. In the revision, we will include the mean±std of $\Delta$ in the main tables.

---

### Official Review · Reviewer_xks2 · 2026-03-07

**Soundness:** 3
**Presentation:** 3
**Significance:** 2
**Originality:** 2
**Overall Recommendation:** 4
**Confidence:** 4

**Summary:**

- This paper focuses not on what to adapt in TTA, but on how to perform adaptation in a stable and effective manner.
- The authors argue that the core issue in TTA lies in the stability-plasticity trade-off: if the learning rate is too large, the model overfits to noisy samples, while if it is too small, adaptation becomes ineffective. To address this, they propose CONGA, a scheduler that adjusts the learning rate using both confidence and gradient information.
- CONGA can be attached to existing TTA methods as a plug-in module, and the paper claims it reduces catastrophic forgetting and noisy gradients while improving performance across diverse benchmarks.

**Compliance With Llm Reviewing Policy:**

Affirmed.

**Final Justification:**

The discussion on Gradient Noise Decomposition appears valid. I am raising my decision to Weak accept.

**Key Questions For Authors:**

* What happens when confidence is high but the gradient is also large? In that case, which gate dominates? Or are the two signals potentially in conflict?

**Limitations:**

see above

**Strengths And Weaknesses:**

Strengths

* The components of the proposed method are well organized and clearly explained, which makes the paper easy to follow.
* The figures effectively summarize the method's main concept, making the control mechanism clear.

Weaknesses

* The authors do not sufficiently discuss or compare against prior work that, very similarly to this paper, adopts a plug-and-play approach, treats TTA from the perspective of the optimization process, and uses a time-varying learning rate strategy [1].
* Prior literature has already suggested that the learning rate is not merely a hyperparameter but a key step-size factor for stable and reliable TTA.
* In particular, the argument for the safety gate, including Theorem 3.2, is very similar to claims already made in prior work.
* There is no component-removal ablation for the cycle regulator, safety gate, and learning gate, so it is difficult to determine whether all of these components are actually necessary.

[1] Lee, Jae-Hong. “Bayesian weight enhancement with steady-state adaptation for test-time adaptation in dynamic environments.” ICML. 2025.

---

> ### Author Rebuttal · Authors · 2026-03-31
>
> **Response to Weaknesses 1,2,3 (comparison with (Lee, 2025)):**
> Thank you for the comments. We acknowledge that both SSA (Lee, 2025) and our method adopt a plug-and-play approach and analyze TTA from the perspective of the optimization process, but their research objective, methodology, and outcome differ:
> 1. The objectives: SSA targets the specific goal of mitigating the effects of explicit noise caused by the unsupervised nature of TTA, while CONGA pursues the objective of maximizing the expected error reduction $\Delta\mathcal{E}_t$​.
> 2. The proposed methodologies: SSA performs steady-state analysis on the distribution of model parameters via SDE approximation and Bayesian filtering, whereas CONGA derives the optimal learning rate by maximizing the lower bound of $\Delta\mathcal{E}_t$.
> 3. The resulting formulations: SSA yields step size $\Delta t_k \propto 1/\sigma_k$​, where $\sigma_k$ denotes the **standard deviation of gradient noise**, while CONGA yields step size $\eta_t^* \propto 1/\sigma_t^2$​, where $\sigma_t$​ denotes the **standard deviation of stochastic noise**. CONGA decomposes gradient noise into a zero-mean stochastic component $\xi_t$​ with variance $\sigma_t^2$​ and a bounded bias term $b_t​$ arising from domain shift, which enables CONGA to selectively suppress stochastic noise while preserving the systematic gradient signal that drives adaptation. Furthermore, the quadratic scaling provides stronger suppression under high stochastic noise, offering more decisive protection against noisy gradient updates.
>
> While prior work has recognized the importance of learning rate in TTA, our work makes the following distinct contributions. First, we provide a systematic analysis of how different LR scheduling strategies affect adaptation stability and performance, which remains unaddressed in existing work. Second, our theoretical framework decomposes gradient noise into stochastic and systematic components, inspiring more principled learning rate design. Third, CONGA operates as a plug-in scheduler applicable to any gradient-based TTA method, combining both gradient magnitude and prediction confidence as complementary signals.
>
> To further understand the relations between CONGA and SSA, we present performance comparison between the two methods using CMF as the baseline. As shown in Table 8 below, our method achieves consistently better results in the tested scenarios. We will continue to conduct complete comparison and discuss the relation between our method and SSA in the revision.
>
> **Table 8: Accuracy (%) comparison of SSA and CONGA on top of CMF (ViT on ImageNet-C).**
> ||Pixelate|JPEG|Brightness|Contrast|Avg.|
> |-|-|-|-|-|-|
> |CMF|75.59|73.11|80.48|68.99|74.54|
> |+SSA|75.52|73.14|80.52|69.34|74.63|
> |+CONGA|75.76|73.55|80.65|69.38|74.83|
>
> **Response to Weakness 4 (component-wise ablation studies)**: Thank you for the comment. We provide a component-wise ablation study in Table 9. As shown, each component contributes positively to the final performance: the Cycle Regulator provides a modest but consistent gain, the Learning Gate further improves accuracy, and the Safety Gate delivers the most significant improvement. These results confirm that all three components are necessary and complementary.
> **Table 9: Component-wise ablation of CONGA with Tent (ViT on ImageNet-Sketch; ResNet on Gaussian Noise).**
> |Components|||ViT||ResNet||
> |:-:|:-:|:-:|:-:|:-:|:-:|:-:|
> |Cycle Regulator|Learning Gate|Safety Gate|Acc.|Gain|Acc.|Gain|
> ||||49.1|-|29.9|-|
> |&#10003;|||49.2|+0.1|30.4|+0.5|
> |&#10003;|&#10003;||49.5|+0.4|30.5|+0.6|
> |&#10003;|&#10003;|&#10003;|50.7|+1.6|34.2|+4.3|
>
>
> **Response to Question (conflict between gates)**: These two signals operate on different bounds and are complementary by design. The gradient magnitude controls $\eta_{\max,t}$ (Safety Gate) while confidence controls $\eta_{\min,t}$ (Learning Gate), so they do not directly compete. Instead, they jointly define the interval within which the cosine schedule operates.
> When confidence is high and gradient is also large, $\eta_{\max,t}$ is reduced while $\eta_{\min,t}$ is raised, narrowing the interval. The cosine schedule still naturally emphasizes $\eta_{\max,t}$ in the early phase and $\eta_{\min,t}$ in the later phase, but within a compressed range that induces more conservative and stable updates overall.
> When this tension is extreme, i.e., $⁡\eta_{\max,t} < \eta_{\min,t}$​, the apparent conflict is in fact a semantically meaningful state indicating a noise-dominated regime where the Safety Gate should take precedence. The inverted interval induces a within-cycle warm-up, providing a principled conservative start before relaxing toward the plasticity floor. EMA smoothing ensures this state is transient, making explicit conflict-resolution logic unnecessary.

---

> > ### Author Rebuttal · Reviewer_xks2 · 2026-04-03
> >
> > Thank you for the detailed rebuttal. The added component-wise ablation in Table 9 and the clarification that the gradient magnitude controls $\eta_{\max,t}$ while confidence controls $\eta_{\min,t}$ are helpful, so some of my concerns are resolved. However, I still have a follow-up question regarding the paper’s novelty relative to SSA. In my original review, I noted that the paper does not “sufficiently discuss or compare against prior work” and that “the argument for the safety gate, including Theorem 3.2, is very similar to claims already made in prior work.” While the rebuttal explains differences in objectives, methodologies, and resulting formulations, and also provides the limited comparison in Table 8, could the authors clarify more explicitly how the Safety Gate and Theorem 3.2 differ from SSA at the theorem/claim level, rather than only at the high-level methodological level?

---

> > > ### Author Response · Authors · 2026-04-04
> > >
> > > Thanks for the constructive follow-up. First, the Safety Gate ($\eta_{\max}$) is independent of SSA. It is inspired by the LARS optimizer and utilizes our proposed logarithmic damping, which effectively stabilizes the modulation trajectory (see Fig. 2). Second, Theorem 3.2 derives the optimal learning rate $\eta_t^* \propto 1/\sigma_t^2$ to provide the theoretical foundation for the Learning Gate ($\eta_{\min}$). While Theorem 3.2 and SSA share some high-level intuition, they diverge fundamentally. We contrast their theoretical foundations as follows:
> > > - **Foundational Assumptions**: SSA, based on steady-state analysis where $P_{k+1} \approx P_k$, implicitly assumes a relatively **static environment** or that the model has nearly **converged**. In contrast, CONGA is built on **non-stationary single-step dynamics**: $e_{t+1} = (1 - \eta_t)e_t - (\eta_t b_t + \delta_t) - \eta_t \xi_t$, where the drift $\delta_t$, stochastic noise $\xi_t$, and systematic bias $b_t$ are all time-varying and highly dynamic. This formulation more faithfully captures the essence of online TTA's real-time updating process compared to the quasi-static assumptions of SSA.
> > > - **Gradient Noise Decomposition**: While both methods utilize gradient noise, we introduce a decomposition of the TTA gradient noise: $k_t = b_t + \xi_t$. Here, $b_t$ represents the systematic bias (with a bounded norm $\|b_t\| \le B$) and $\xi_t$ is the zero-mean stochastic noise with variance $\mathbb{E}[\|\xi_t\|^2] = \sigma_t^2$. We prove that the optimal learning rate is dominated by $1/\sigma_t^2$, allowing for the selective suppression of stochastic noise while preserving the directional signal of $b_t$ for effective domain adaptation. Conversely, SSA treats the standard deviation $\sigma_k$ of gradient noise as a monolithic interference term, failing to distinguish between these two sources. Consequently, SSA may simultaneously suppress both random noise and the beneficial domain-shift signal. This fundamental difference in decomposition leads to our second-order suppression ($1/\sigma_t^2$), which more aggressively mitigates isolated stochastic noise, whereas SSA relies on a first-order approach ($1/\sigma_k$) to avoid losing bias information that it cannot explicitly disentangle from noise.
> > >
> > > We will discuss this further in the revision.

---

### Official Review · Reviewer_8mrQ · 2026-03-12

**Soundness:** 3
**Presentation:** 2
**Significance:** 3
**Originality:** 3
**Overall Recommendation:** 4
**Confidence:** 3

**Summary:**

This paper investigates test-time adaptation (TTA) through the lens of learning rate scheduling and proposes CONGA. This plug-and-play scheduler can be incorporated into existing TTA methods without modifying the model architecture or loss function. Motivated by a theoretical analysis of the trade-off between error contraction and gradient noise during adaptation, CONGA dynamically adjusts the learning rate to improve adaptation under distribution shift.

**Compliance With Llm Reviewing Policy:**

Affirmed.

**Final Justification:**

My concerns have been addressed.

**Key Questions For Authors:**

Due to my limited expertise, my review may contain misunderstandings or inaccuracies. Please feel free to point them out and correct them.

**Limitations:**

yes

**Strengths And Weaknesses:**

**Strengths**
1. This paper studies TTA from the more fundamental perspective of learning rate scheduling, offering a new research viewpoint.
2. By analyzing error tracking, it reveals the trade-off between error contraction and gradient noise in TTA, providing theoretical motivation for dynamically adjusting the learning rate.
3. As a scheduler, CONGA can be directly integrated with existing TTA methods without modifying the model architecture or loss function.

**Weaknesses**
1. The trigger conditions of Safety Guard (3× base LR, confidence 0.75) and the confidence threshold of Learning Gate (0.9) are all tunable hyperparameters, which is inconsistent with the paper’s claim that CONGA is “parameter-free.”
2. As shown in Table 7, ResNet uses $\eta = 2.5 \times 10^{-4}$, while CONGA uses $\eta_{\text{base}} = 2.0 \times 10^{-3}$. This suggests that the performance improvement may mainly come from the larger initial learning rate, rather than from the CONGA scheduling mechanism itself.
3. Table 8 shows that Tent has a standard deviation as high as 13.6 in the Fog category, while the authors’ method reaches 25.6. These experimental results seem quite unusual.
4. The paper assumes that $1/\sigma_t^2 \propto C_t^2$, and the cited justification comes from the importance sampling literature in supervised learning. Does this assumption still hold in the unsupervised TTA setting?
5. The proof of Theorem 3.1 is based on gradient updates under a local quadratic loss $f_t(\theta)=\frac{1}{2}\|\theta-w_t\|^2$, whereas TTA in practice usually uses an entropy minimization loss. Can the theoretical conclusions be generalized to the practical setting?

---

> ### Author Rebuttal · Authors · 2026-03-31
>
> **Response to Weakness 1 (parameter-free)**:
> Thanks for the comment. We first clarify that "parameter-free" in our paper refers to the absence of additional trainable parameters, which we will revise as "train-free" to avoid ambiguity. Regarding the sensitivity of Safe Guard triggers and confidence threshold, all experiments presented in the paper use a unified set of hyperparameters. Table 5 below presents the sensitivity analysis for the Safe Guard, and Table 3 in response to Reviewer G1Bz covers the confidence threshold. Both results demonstrate CONGA's robustness to these choices. This conclusion is generalizable to other experiment settings.
> **Table 5: Sensitivity to Safe Guard with DeYO on ImageNet-Sketch (ViT).**
> ||Base LR Upper|Conf. Upper|Acc.|
> |-|-|-|-|
> |Highly Strict|1.5*base_lr|0.6|52.0|
> |Strict|2*base_lr|0.7|52.7|
> |Moderate (selected)|3*base_lr|0.75|53.1|
> |Loose|4*base_lr|0.8|53.2|
> |Very Loose|5*base_lr|0.9|53.0|
>
> **Response to Weakness 2 (choice of initial learning rate)**:
> As shown in Table 6 below, simply increasing the baseline learning rate to 2e-3 does not yield consistent improvements. Instead, the performance drops by 7% on EATA, and methods sensitive to large learning rates (e.g., DeYO) may even collapse. Furthermore, due to the decay nature of CONGA, the average effective learning rate throughout our adaptation process is close to 2.5e-4. These evidences suggest that the performance gains of CONGA do not stem from a larger learning rate.
> **Table 6: Performance comparison on ResNet using $\eta$=2e-3, where values in parentheses denote the accuracy changes relative to $\eta$=2.5e-4.**
> ||Gaussian Noise|Shot Noise|Impulse Noise|Avg.|
> |-|-|-|-|-|
> |EATA|26.7(-8.9)|31.0(-6.4)|30.9(-5.7)|29.5(-7.0)|
> |Tent|30.2(+0.3)|32.4(+0.4)|31.5(+0.2)|31.4(+0.3)|
>
>
> **Response to Weakness 3 (unusual standard deviation)**:
> Thanks for the comment. We acknowledge the transcription error in current manuscript, while the actual standard deviation of Tent on the Fog category is even higher. The raw results across 5 random seeds are provided in Table 7.
> **Table 7: Results of different seed in the Fog category with Tent on ViT.**
>
> ||seed=2019|2020|2021|2022|2023|
> |-|-|-|-|-|-|
> |Tent|2.4|3.1|2.4|2.2|73.0|
> |+Ours|73.1|72.2|2.4|2.2|73.2|
>
> As shown, Tent converges normally under only 1 out of 5 seeds, collapsing in the remaining 4. Such result is consistent with the near-random accuracy (~2.3%) reported for Tent on Fog in prior works [1][2]. With CONGA, we are able to reduce the probability of collapse, demonstrating that CONGA improves convergence stability under this challenging corruption. We will correct the results and provide additional analysis of this phenomenon in the revision.
> [1]Niu, S., Miao, C., Chen, G., Wu, P., and Zhao, P. Test-Time Model Adaptation with Only Forward Passes. ICML 2024.
> [2]Deng, Q., Niu, S., Zhang, R., Chen, Y., Zeng, R., Chen, J., and Hu, X. Learning to Generate Gradients for Test-Time Adaptation via Test-Time Training Layers. AAAI 2025.
>
> **Response to Weakness 4 (Validity of $1/\sigma_t^2 \propto C_t^2$ in TTA)**:
> Thanks for the question. We clarify that we do not adopt the sample selection methodology from the cited work, but only the fundamental mathematical relationship: that gradient noise variance is inversely proportional to prediction confidence, which is a property of the model's loss landscape and does not depend on the ground-truth labels. In the unsupervised TTA setting, samples with high confidence are likely correct and the gradient estimate is approximately unbiased, recovering the supervised case where $1/\sigma_t^2\propto C_t^2$ holds. When confidence is low, CONGA responds by lowering $\eta_{min, t}$, reducing the effective learning rate and limiting the influence of such samples by design. Therefore, the assumption naturally holds in the unsupervised TTA setting.
>
> **Response to Weakness 5 (Validity of $f_t(\theta) = \frac{1}{2}\|\theta - w_t\|^2$ in TTA)**:
> Thanks for the question. The quadratic surrogate $f_t(\theta) = \frac{1}{2}\|\theta - w_t\|^2$ approximates any twice-differentiable loss locally via second-order Taylor expansion:
> > $$\mathcal{L}(\theta) \approx \mathcal{L}(\theta_t) + \nabla \mathcal{L}(\theta_t)^\top (\theta - \theta_t) + \frac{1}{2}(\theta - \theta_t)^\top H_t (\theta - \theta_t)$$
> Entropy minimization $\mathcal{L} = -\sum_k p_k \log p_k$ is twice-differentiable on the simplex interior, so the quadratic approximation is valid in the high-confidence regime where the model parameters change slowly and the loss landscape is flat, making the quadratic approximation accurate within the small update steps taken by gradient descent.
> In the low-confidence regime where differentiability may break down for some classes, our adaptive mechanism suppresses gradient updates, limiting the practical impact of any approximation gap. Therefore, the conclusions of Theorem 3.1 remain valid in this setting.

---

> > ### Author Rebuttal · Reviewer_8mrQ · 2026-04-03
> >
> > Thank you to the authors for the response. I will maintain my score.

---

> > > ### Author Response · Authors · 2026-04-06
> > >
> > > Thanks for your recognition and for the positive feedback. We truly appreciate your recognition of our efforts and your valuable comments, which have helped us further improve the paper.

---

### Official Review · Reviewer_G1Bz · 2026-03-12

**Soundness:** 2
**Presentation:** 3
**Significance:** 3
**Originality:** 3
**Overall Recommendation:** 4
**Confidence:** 5

**Summary:**

This paper proposes CONGA, a confidence- and gradient-aware learning rate scheduler designed for test-time adaptation (TTA). The key idea is to dynamically adjust the learning rate based on the ratio between gradient magnitude and parameter magnitude, together with prediction confidence, in order to achieve a safer and more stable adaptation process. The method also introduces mechanisms such as a learning gate and periodic restarts to control unstable updates during adaptation. Extensive experiments on several standard TTA and continual TTA benchmarks demonstrate that CONGA can improve performance and stability compared to existing adaptation strategies.

**Compliance With Llm Reviewing Policy:**

Affirmed.

**Final Justification:**

The rebuttal addresses my concerns. I maintain my original positive score.

**Key Questions For Authors:**

1. The experiments in the paper mainly focus on standard TTA or CTTA settings, which usually assume relatively balanced class distributions and moderate batch sizes. Could the authors evaluate the method under more challenging scenarios, such as label shifts or extremely small batch sizes (e.g., BS=1)?
2. Is a fixed restart period (e.g., 100 iterations) optimal across different settings? For instance, if a domain shift occurs around iteration 50 but the restart happens at iteration 100, how does the method handle the potential performance degradation during this interval?

**Limitations:**

1. Several hyperparameters, including the restart period T_i, the EMA coefficient alpha, and the Safe Guard threshold, are manually specified. This introduces additional tuning overhead in practical applications, and the method is still somewhat far from being parameter-free.
2. Since the method requires backpropagation to compute gradient statistics, it may be less suitable for deployment on devices that only support forward inference. In such cases, CONGA may be less practical compared to some backpropagation-free adaptation methods.

**Strengths And Weaknesses:**

**Strengths:**
1. CONGA is conceptually simple and can be integrated into existing TTA pipelines with minimal modifications, making it a practical plug-in component for adaptive learning rate control.
2. The paper focuses on the problem of learning rate stability during test-time adaptation and proposes an adaptive scheduling strategy to mitigate unstable updates. This perspective is meaningful since learning rate selection is a critical yet often overlooked factor affecting the robustness of TTA methods.

**Weaknesses:**
1. Since CONGA aims to learn a safe adaptive learning rate during test-time adaptation, it would be important to evaluate its robustness to different base learning rate choices. However, the current experiments use a fixed base learning rate without providing a sensitivity analysis. Evaluating the method under multiple base LR settings would help verify whether the proposed scheduler can consistently maintain stable performance across different datasets and adaptation scenarios.
2. The core mechanism of CONGA relies on the ratio between the gradient magnitude and the parameter magnitude computed from the current batch, using it as a proxy for gradient noise or the aggressiveness of parameter updates. However, such a design may introduce a certain degree of short-sightedness. In practical scenarios, if the current batch contains significant noise, outliers, or label shifts, the resulting gradients may not accurately reflect the true optimization direction. This could lead the learning gate to make suboptimal step-size adjustments.
3. The paper does not provide a mechanism for detecting domain shifts and dynamically triggering restarts. Although Figure 4 compares different values of T_i in realistic Continual Test-Time Adaptation (CTTA) scenarios the frequency of domain shifts is typically unknown. Therefore, relying on a fixed restart schedule may limit the method’s adaptability when the timing of distribution changes varies.
4. The method introduces a confidence threshold C_t as part of the safe learning rate control mechanism. However, the paper does not provide a sensitivity analysis for this hyperparameter. It would be helpful to evaluate how different choices of C_t affect the stability and performance of the method.

---

> ### Author Rebuttal · Authors · 2026-03-31
>
> **Response to Weakness 1 (based learning rate sensitivity analysis):**
> As shown in Table 1, CONGA uniformly improves over both baselines across all tested base learning rates (LR). Moreover, the performance of Tent ranges from 51.9% to 54.2% with varying base LR, indicating a 2.3% accuracy fluctuation. CONGA narrows such fluctuation to 1.8%, indicating more stable performance across different base LR choices. A similar conclusion can also be observed from DeYO, where CONGA reduces the accuracy fluctuation from 1.3% to 0.9%.
> **Table 1: Sensitivity analysis for different base LR on ImageNet-A (ViT).**
> ||1e-3 (selected)|2e-3|3e-3|9e-4|8e-4|5e-4|
> |-|-|-|-|-|-|-|
> |Tent|52.9|53|54.2|52.3|52.7|51.9|
> |+Ours|53.6|54.5|54.5|54.0|53.7|52.7|
> |DeYO|55.2|55.2|55.3|55.2|55.2|54.0|
> |+Ours|55.5|55.2|55.4|55.8|55.5|54.9|
>
> **Response to Weakness 2 (short-sightedness of CONGA):**
> Thanks for the comment. We would like to highlight that CONGA incorporates two mechanisms to mitigate this concern. First, the Safety Gate $\eta_{\max,t}$​ is computed from the EMA of parameter and gradient norms, ensuring that step-size modulation reflects long-term optimization dynamics rather than single-batch noise. Second, the proposed logarithmic damping suppresses the impact of outlier gradients common in unstable TTA environments. As evidenced in Fig. 2 in the paper, this design maintains LR stability. Therefore, CONGA is robust against sudden noises from shifts in certain batches.
>
> **Response to Weakness 3 (fixed vs. dynamic restarts):**
> To investigate the effects of dynamic and our static restarts, we evaluate CONGA integrated with Reservoir (a domain-detection method for CTTA) across two realistic CTTA scenarios.
> - Evaluated CONGA variants: *global CONGA* adopts a fixed restart length of 100 regardless of detected domain changes, as in the paper. *per-domain CONGA* maintains an LR schedule for each domain detected by Reservoir. The LR schedule for different domains evolves and restarts independently, thus eliminating the influence of domain shifts during restarts.
> - Evaluated CTTA scenarios: *CTTA* is the setting in our paper where domains change smoothly. *CTTA+CDC* adopts the Continual Dynamic Change (CDC) setting from Reservoir, bringing abrupt and unpredictable distribution shifts.
>
> **Table 2: Accuracy on ImageNet-C under CTTA and CTTA+CDC scenarios (ViT). Columns represent recurring domain visits.**
> |scenario|baseline|2|4|6|8|10|Avg.|
> |-|-|-|-|-|-|-|-|
> |CTTA|reservoir|62.8|64.4|65.2|65.7|66.1|64.4|
> ||+global CONGA|63.9|65.5|66.0|66.5|66.6|65.3|
> ||+per-domain CONGA|64.0|65.3|66.0|66.4|66.7|65.3|
> |CTTA+CDC|reservoir|59.0|60.4|61.0|61.7|62.2|60.5|
> ||+global CONGA|59.5|64.7|65.5|66.0|66.2|64.0|
> ||+per-domain CONGA|62.4|64.6|65.4|65.9|66.2|64.2|
>
> As shown in Table 2, the two variants achieve comparable overall results across both scenarios. Although global CONGA falls behind in initial stages of CTTA+CDC due to EMA warm-up, the final results remain stable because the periodic high-LR restarts in CONGA's cosine schedule provide sufficient adaptation capacity to recover from the initial misalignment. This mechanism bridges the performance gap introduced by the EMA warm-up within a few visits.
>
> **Response to Weakness 4 ($C_t$ sensitivity)**:
> Thanks for the suggestion. We conduct a sensitivity analysis on $C_t$ as shown in Table 3. All tested $C_t$ values outperform the baseline (DeYO with constant LR) and the overall accuracy is stable within the tested range. This indicates that CONGA is robust to different choices of $C_t$.
> **Table 3: Sensitivity of $C_t$ with DeYO on ImageNet-Sketch (ViT).**
> ||baseline|0.95|0.9 (selected)|0.85|0.8|0.75|
> |:-:|:-:|:-:|:-:|:-:|:-:|:-:|
> |Acc.|52.2|53.0|53.0|52.6|53.2|52.8|
>
>
> **Response to Question 1 (more challenging scenarios)**:
> As shown in Table 2 above, CONGA achieves a 3.7% improvement (60.5→64.2) in the more challenging CTTA+CDC scenario, demonstrating CONGA's effectiveness under harder shift conditions. Table 4 below further reports results under batch-size-1 setting, where CONGA consistently surpasses the baselines (+6% on average for TENT and +2.8% for EATA), confirming its effectiveness in the single-sample adaptation setting.
> **Table 4: Accuracy with batch size=1 on Tent (ViT) and EATA (ResNet).**
> ||Pixelate|Elastic|Contrast|Avg.|
> |-|-|-|-|-|
> |EATA|46.5|27.0|44.5|39.3|
> |+Ours|51.2|27.9|47.1|42.1|
> |Tent|67.0|54.0|65.0|62.0|
> |+Ours|73.6|62.6|67.7|68.0|
>
> **Response to Question 2 (shift-restart mismatch)**:
> Thanks for the question. The CTTA+CDC scenario in Table 2 directly addresses this concern. Domain shifts in this setting occur randomly and misalign with the fixed restart cycle. The results show that global CONGA achieves comparable final performance with dynamic restarts. As analyzed above, we attribute this to CONGA's periodic high-LR restarts, which serve as an implicit error-correction mechanism for cycle misalignment.

---

> > ### Author Rebuttal · Reviewer_G1Bz · 2026-04-04
> >
> > - W1 : Resolved — new Table 1 across six LR values shows consistent improvements and reduced variance.
> > - W2: Partially Resolved — EMA and log-damping arguments are sound, but no direct ablation under corrupted-batch conditions was provided.
> > - W3 : Resolved — CTTA+CDC experiments show global and per-domain CONGA converge to nearly identical averages.
> > - W4 : Resolved — sensitivity table demonstrates <0.6% variation across tested C_t values.
> > - Q1 : Resolved — BS=1 results (+6% on Tent) and CTTA+CDC (+3.7%) are compelling.
> > - Q2 : Resolved — addressed jointly with W3 via CTTA+CDC experiments.
> >
> > Overall, the rebuttal is substantive and well-organised. The authors provide new experiments that directly address four of my five concerns. The one remaining gap (W2) is a minor empirical gap rather than a fundamental flaw — the theoretical justification for EMA and log-damping is sound, and the practical evidence (Fig. 2) is supportive even if not comprehensive. The paper's core contribution — framing TTA from the optimization scheduling perspective and demonstrating consistent, lightweight improvements — is meaningful and the rebuttal has strengthened my confidence in its soundness.
> >
> > I intend to maintain my score on the positive side.

---

> > > ### Author Response · Authors · 2026-04-06
> > >
> > > We sincerely appreciate your time in reading our rebuttal and your decision to maintain a positive rating. Your supportive feedback is truly encouraging to us.

---

### Decision · Program_Chairs · 2026-04-30

**Decision:**

Accept (regular)

**Comment:**

All four reviewers gave the rating 4 (Weak Accept). The reviewers generally agree that the paper is technically sound, well-motivated, and practically useful, offering a meaningful perspective by focusing on optimization stability rather than architectural changes. The method is simple to integrate and demonstrates consistent empirical improvements across multiple TTA and CTTA benchmarks, which is viewed as a key strength.

The rebuttal is strong and comprehensive, addressing most reviewer concerns with additional experiments and clarifications, including sensitivity analyses (base learning rate, thresholds), challenging settings (batch size 1, dynamic shifts), component-wise ablations, and statistical reporting (mean $\pm$ std). As a result, multiple reviewers explicitly acknowledge that their concerns have been resolved and maintain or raise their positive scores.

Overall, I would like to recommend accept.

---

I guess this error was caused by a mistake and did not report it to the PC chair. Please fix the author mismatch.

Automatically flagged references

Reference: Adam, K. D. B. J. et al. A method for stochastic optimization. arXiv preprint arXiv:1412.6980, 1412(6), 2014.
Issue: authors mismatch with arXiv